# Conformal Mixed-Integer Constraint Learning with Feasibility Guarantees

**Daniel Ovalle**
Department of Chemical Engineering
Carnegie Mellon University
dovallev@andrew.cmu.edu

**Lorenz T. Biegler**
Department of Chemical Engineering
Carnegie Mellon University
lb01@andrew.cmu.edu

**Ignacio E. Grossmann**
Department of Chemical Engineering
Carnegie Mellon University
ig0c@andrew.cmu.edu

**Carl D. Laird**
Department of Chemical Engineering
Carnegie Mellon University
claird@andrew.cmu.edu

**Mateo Dulce Rubio**
Center for Data Science
New York University
mateo.d@nyu.edu

## Abstract

We propose Conformal Mixed-Integer Constraint Learning (C-MICL), a novel framework that provides probabilistic feasibility guarantees for data-driven constraints in optimization problems. While standard Mixed-Integer Constraint Learning methods often violate the true constraints due to model error or data limitations, our C-MICL approach leverages conformal prediction to ensure feasible solutions are ground-truth feasible with probability at least $1-\alpha$, under a conditional independence assumption. The proposed framework supports both regression and classification tasks without requiring access to the true constraint function, while avoiding the scalability issues associated with ensemble-based heuristics. Experiments on real-world applications demonstrate that C-MICL consistently achieves target feasibility rates, maintains competitive objective performance, and significantly reduces computational cost compared to existing methods. Our work bridges mathematical optimization and machine learning, offering a principled approach to incorporate uncertainty-aware constraints into decision-making with rigorous statistical guarantees.

## 1 Introduction

Constraint Learning (CL) deals with inferring the functional form of constraints or objectives from observed data to embed them into mathematical optimization problems, particularly when parts of the system behavior are difficult to model explicitly [1]. This is especially relevant in settings where the relationship between decision and output variables is unknown but can be learned from data, enabling a supervised learning approach [2, 3]. It is also valuable in surrogate modeling scenarios where the relationship is known but difficult to optimize, either due to its nonlinearity, computational cost, or complexity [4–6]. That is, CL simply leverages modern machine learning systems to learn the structure of complex constraints directly from data, enabling prediction of constraint boundaries at unobserved points to efficiently navigate feasible spaces and identify optimal solutions.

39th Conference on Neural Information Processing Systems (NeurIPS 2025).

The core principle that enables CL is that the predictions of many machine learning models can be formulated as closed-form algebraic expressions that are optimization-compatible and commonly admit exact mixed-integer programming (MIP) representations [1, 7–17]. This integration is commonly referred to as *Mixed-Integer Constraint Learning* (MICL). However, a critical challenge arises from the very nature of optimization algorithms: they systematically leverage the mathematical structure of the problem to identify extreme points. As a result, small approximation errors in the learned constraint model (e.g., arising from noisy data, model bias, or insufficient training) can be systematically exploited during the optimization. This behavior often leads to solutions that are infeasible with respect to the true system, meaning that the optimal decision variables found by optimization programs under CL frequently violate the original constraints when evaluated using the true underlying model [18]. This lack of *true* feasibility results in solutions that cannot be implemented in practice, or that are overly conservative due to the need for additional safety margins.

To improve the robustness of MICL solutions, recent research has proposed ensemble-based approaches where multiple predictive models are trained on different bootstrap samples of the data [19]. In particular, Maragno et al. [7] introduce a framework in which $P$ models are independently trained on bootstrapped datasets, and the optimization is constrained so that a $1-\alpha$ fraction of these models must satisfy the imposed constraints. There is some empirical evidence that suggests that this heuristic yields solutions that are typically feasible with respect to the true underlying system, and its robustness improves as $P$ increases. However, this approach lacks formal guarantees as there is no assurance that the final solution will satisfy the true constraints with high probability, particularly in the presence of model misspecification or low-quality data used in the MICL step. Moreover, the method suffers from key scalability issues: as $P$ grows, the size of the optimization problem increases accordingly due to the replication of model constraints and auxiliary variables, rapidly making the problem computationally intractable for large ensembles.

In this paper, we introduce *Conformal Mixed-Integer Constraint Learning* (C-MICL), a novel model-agnostic framework for incorporating data-driven constraints into optimization problems with formal probabilistic feasibility guarantees. Our method supports both regression and classification settings and is computationally efficient without sacrificing significant solution quality. Specifically:

- **Probabilistic Guarantees:** We provide a new optimization formulation that guarantees ground-truth feasibility with probability at least $1-\alpha$ when the true constraint function is unknown or inaccessible, under a mild conditional independence assumption.

- **Model-Agnostic and MIP-Compatible:** C-MICL is compatible with any predictive model that admits a MIP encoding, enabling integration into standard optimization solvers.

- **Scalability and Efficiency:** Our method avoids query access or model refinement during optimization, does not scale with dataset size, and requires training at most two models, achieving orders-of-magnitude computational speedup over ensemble-based heuristics.

- **Empirical Validation:** We demonstrate the effectiveness of C-MICL on real-world case studies in both regression and classification settings. Extensive experiments show that our method consistently achieves target feasibility levels, while delivering competitive objective values with significantly reduced computation times compared to state-of-the-art baselines.

## 2 Literature Review

**Mixed-Integer Constraint Learning.** The mixed-integer constraint learning framework has enabled a wide range of developments in machine learning, including neural network verification [20–27], adversarial example generation [28–30], and applications in reinforcement learning [31–33]. It has also been successfully applied to domain-specific tasks in power systems [34, 35], healthcare [36, 37], supply chain management [38, 39], and chemical process design [40]. For a comprehensive overview of recent advances in constraint learning methodologies, we refer readers to Fajemisin et al. [1], and for a review focused on engineering applications, see Misener and Biegler [41]. However, existing CL frameworks generally lack guarantees of true system feasibility, particularly in settings where predictive uncertainty from noisy data or model misspecification is prevalent but typically ignored. The approach proposed in this work directly addresses this gap by providing formal probabilistic feasibility guarantees for uncertain learned constraints embedded into optimization problems.

**Trust-Region-Filter for Optimization.** Trust region filter methods have been proposed as a strategy to optimize over learned constraints while ensuring that the solution aligns with that of the true, underlying system [18]. These approaches operate by iteratively refining the learned constraint through local sampling and evaluation of the truth function, progressively narrowing the optimization domain within trust regions where the learned model is accurate [42, 43]. However, such methods rely on strong assumptions: they typically assume noise-free data, differentiability of the learned constraint, and direct query access to the truth function. In contrast, our framework is designed for settings where these assumptions do not hold. We account for noisy data, allow for non-differentiable learned models, and, most importantly, do not require access to or knowledge of the underlying function, enabling optimization with high-probability feasibility guarantees even under limited and imperfect information.

**Conformal-Enhanced Optimization.** Building on advances in conformal prediction [44–46], recent work has explored its integration with optimization. Lin et al. [47] proposed conformalized inverse optimization, targeting the estimation of unknown parameters embedded in the objective or constraints of an optimization problem, and providing uncertainty quantification over these learned parameters. Our approach addresses a different problem, namely learning and conformalizing the functional forms of constraints themselves, without relying on a fixed parametric structure, which allows for more expressive modeling of complex or unknown system behaviors. Separately, Zhao et al. [48] introduced conformalized chance constrained optimization, in which constraints involving random variables are required to hold with high probability. Their method encodes quantile thresholds from conformal prediction directly into the optimization formulation, which scales with the number of data points. Unlike this approach, ours is focused on uncertainty of model predictions rather than model stochasticity, and supports learning general constraint functions.

Other related efforts include Johnstone and Cox [49], who construct conformal ellipsoids for robust optimization, and Yeh et al. [50], who learn convex uncertainty sets for robust objectives using differentiable conformal layers. In contrast, C-MICL conformalizes feasibility regions rather than robustifying objective values. Similarly, Kiyani et al. [51] and Patel et al. [52] employ conformal sets for utility-aware or risk-sensitive decision-making, optimizing value-at-risk or ensuring robust output quality, whereas C-MICL focuses on constraint satisfaction under uncertainty from model estimation. Our framework supports general, nonconvex feasible regions and provides distribution-free probabilistic feasibility guarantees from fixed-size optimization formulations regardless of dataset size, allowing practitioners to leverage large datasets for improved model accuracy without inflating computational cost. A key distinction of C-MICL is that it centers on feasibility-aware decision-making rather than on robustifying objectives or actions for downstream optimization problems.

## 3 Preliminaries

### 3.1 Problem Set-Up

Consider a constrained optimization problem over a set of decision variables $(x, y, z)$ where certain constraints depend on unknown or difficult-to-model functions relating decision variables $x$ to outcomes $y$. We assume no access to the true underlying mapping denoted by $h(x) = \mathbb{E}[Y \mid X = x]$. Instead, we have access to a dataset $\mathcal{D}_{\text{train}} = \{(x_i, y_i)\}_{i=1}^{N_{\text{train}}}$, where each observation $(x_i, y_i)$ is drawn independently from a common (unknown) underlying distribution $\mathcal{P}_{XY}$. Note that $z$ are additional decision variables that may constrain $x$, but do not influence $y$ through $h(x)$.

The goal of Constraint Learning is to approximate $h(x)$ with a predictive model $\hat{h}(x)$ trained on the observed data $\mathcal{D}_{\text{train}}$. Once the model $\hat{h}(x)$ is trained, we embed it in a mixed-integer optimization program, defining the following Mixed-Integer Constraint Learning (MICL) problem [1]:

$$
\begin{aligned}
\min_{x,y,z} \quad & f(x, z) \\
\text{s.t.} \quad & g(x, z) \leq 0 \\
& \hat{h}(x) = y \\
& (x, z) \in \mathcal{X} \\
& y \in \mathcal{Y}
\end{aligned}
\tag{MICL}
$$

where $f(x, z)$ is a known real-valued objective function and $g(x, z)$ encodes a set of known constraints. Moreover, $\mathcal{X} \subseteq \mathbb{R}^{m_1} \times \mathbb{Z}^{m_2}$ defines the feasibility set for the decision variables $(x, z)$ where both $x$ and $z$ can take both real and integer values. Finally, $\mathcal{Y}$ defines the feasible set for the outcome variable $y$, where feasibility must be satisfied with respect to the true (but unknown) function $h(x)$.

The formulation above captures a general form of a mixed-integer constraint learning problem. Specific problem instances correspond to different definitions of the feasible prediction set $\mathcal{Y}$. In the regression setting, $\mathcal{Y}$ is any subset of $\mathbb{R}$ that is representable within a MIP framework. For instance, in our experiments, we define $\mathcal{Y} = [\underline{y}, \overline{y}]$, which enforces lower and upper bounds on the continuous output variable $y$.[1] In the multi-label classification case, $\mathcal{Y} = \mathcal{K}^{\text{des}}$ is a subset of the set of possible labels $\mathcal{K} = \mathcal{K}^{\text{des}} \cup \mathcal{K}^{\text{und}}$, which we assume partitions into disjoint sets of *desired* classes $\mathcal{K}^{\text{des}}$ and *undesired* classes $\mathcal{K}^{\text{und}}$.

By construction, all feasible solutions $(x', z')$ from the optimization problem MICL are feasible with respect to the learned model $\hat{h}(x)$, meaning that $\hat{h}(x') \in \mathcal{Y}$. However, since $h(x)$ is unknown and only approximated via $\hat{h}(x)$, *ground-truth feasibility* with respect to the true function $h(x')$ is not guaranteed. We formalize this conceptual difference in the following definition.

**Definition 3.1.** *(Ground-Truth Feasibility) Given a feasible solution $(x', z')$ to the MICL optimization problem, the solution is said to achieve ground-truth feasibility if $y' = h(x') \in \mathcal{Y}$, where $h(x)$ is the true constraint function.*

Recent research by Maragno, Wiberg, Bertsimas, Birbil, den Hertog, and Fajemisin [7] proposes a heuristic approach called Wrapped model MICL (W-MICL) to address the challenge of ensuring ground-truth feasibility of MICL solutions. W-MICL trains an ensemble of $P$ models using independent bootstrapped datasets rather than relying on a single predictive model. Each model $\hat{h}_p(x)$ provides a distinct prediction, collectively capturing model uncertainty. Within the MICL optimization formulation, the constraint $\hat{h}(x) \in \mathcal{Y}$ is replicated for each trained model, and feasibility $\hat{h}_p(x) \in \mathcal{Y}$ is enforced for a $(1 - \alpha)$ fraction of them, implemented via standard big-M formulations in MIP [7]. While empirical results suggest that W-MICL improves the ground-truth feasibility of optimal solutions, the method remains a heuristic and does not offer formal feasibility guarantees.

**Remark 3.1.** *We assume that the underlying predictive model $\hat{h}(x)$ admits a reformulation as a mixed-integer program (MIP), such as Neural Networks with ReLU activations (ReLU NNs) [8–10], Gradient-Boosted Trees (GBTs) [11, 12], Random Forests (RFs) [13–16], or Linear-Model Decision Trees (LMDTs) [17]. Then, the MICL problem is itself a MIP that can be solved using standard branch-and-bound or global optimization algorithms [53, 54]. While details on embedding these models into MIPs are provided in Appendix A, we emphasize that our formulation remains model-agnostic, requiring only that the predictive model $\hat{h}(x)$ is MIP-representable.*

### 3.2 Conformal Prediction

Conformal prediction is a statistical framework for model-free uncertainty quantification that provides probabilistic guarantees for prediction sets constructed from any given (fixed) predictive model. This framework has gained significant popularity as an assumption-lean, model-agnostic, computationally efficient method offering rigorous statistical coverage guarantees without requiring model retraining or modifications to pre-trained machine learning systems [55–57].

Formally, suppose we have access to a *calibration dataset* of $N$ data points $(x_1, y_1), \ldots, (x_N, y_N)$ drawn exchangeably from the underlying distribution $\mathcal{P}_{XY}$.[2] Given a new data point $(x_{N+1}, y_{N+1})$ drawn exchangeably from $\mathcal{P}_{XY}$, conformal prediction provides a principled method for constructing a *conformal set* $\mathcal{C}_\alpha(x_{N+1})$ containing the *true* label $y_{N+1}$ with probability at least $1 - \alpha$, without making additional assumptions about the underlying distribution beyond exchangeability.

The core idea behind conformal prediction is to compute *conformal scores* $\{s(x_i, y_i)\}_{i=1}^N$, that capture model uncertainty over the calibration data. For instance, in regression settings, a common *conformal score function* corresponds to model absolute residuals $s(x, y) = |y - \hat{h}(x)|$. The

---

[1]If $y$ is a vector-valued outcome, $[\underline{y}, \overline{y}]$ imposes elementwise constraints on $y \in \mathcal{Y}$.

[2]A random vector $(Z_1, \ldots, Z_N)$ is exchangeable if $(Z_{\sigma(1)}, \ldots, Z_{\sigma(N)})$ follows the same distribution for all permutations $\sigma$ of the indices $\{1, \ldots, N\}$ [57]. For instance, *i.i.d.* random variables are exchangeable.

conformal prediction set is then defined as:

$$\mathcal{C}_\alpha(x_{N+1}) = \{y : s(x_{N+1}, y) \le \widehat{q}_{1-\alpha}\} \tag{1}$$

where $\widehat{q}_{1-\alpha}$ is the $(1-\alpha)(1 + 1/N)$-quantile of the empirical distribution of conformal scores $\{s(x_1, y_1), \ldots, s(x_N, y_N)\}$. This construction yields the following key marginal guarantee.

**Theorem 3.1** (Theorem 3.2 in [57])**.** *Assume that* $(X_1, Y_1), \ldots, (X_{N+1}, Y_{N+1})$ *are exchangeable random variables. Then, for any user-specified coverage level* $\alpha \in (0, 1)$*, we have that the conformal set from eq.* (1) *satisfies*

$$\mathbb{P}\left(Y_{N+1} \in \mathcal{C}_\alpha(X_{N+1})\right) \ge 1 - \alpha. \tag{2}$$

Recent advancements have extended the framework to construct *adaptive* prediction sets with conditional coverage guarantees, whose size varies according to the model's confidence at specific inputs, producing narrower intervals in regions of low uncertainty [58, 59]. This adaptive property is particularly relevant to our work, as it informs optimization algorithms about local prediction confidence when checking for feasibility. For instance, *Mondrian conformal prediction* [57, 60] enables conditional coverage guarantees for both ground-truth feasible points $\{x : h(x) \in \mathcal{Y}\}$ and infeasible points $\{x : h(x) \notin \mathcal{Y}\}$. This conditional coverage guarantee is formalized in the following Lemma, with the proof and details on Mondrian conformal sets deferred to Appendix B.1.

**Lemma 3.1.** *(Ground-Truth Feasibility Conformal Coverage Guarantee): Assume that* $(X_1, Y_1), \ldots, (X_{N+1}, Y_{N+1})$ *are exchangeable random variables and that* $s(X, Y)$ *is a symmetric score function. Mondrian conformal sets satisfy the conditional coverage guarantee at level* $\alpha$*:*

$$\mathbb{P}(Y_{N+1} \in \mathcal{C}_\alpha(X_{N+1}) \mid Y_{N+1} \in \mathcal{Y}) \ge 1 - \alpha,$$
$$\mathbb{P}(Y_{N+1} \in \mathcal{C}_\alpha(X_{N+1}) \mid Y_{N+1} \notin \mathcal{Y}) \ge 1 - \alpha.$$

Intuitively, this condition ensures that coverage holds simultaneously within both the feasible and infeasible regions, by constructing conformal sets calibrated separately for each group. This conditional guarantee is strictly stronger than (and implies) the marginal property from Theorem 3.1, preventing the procedure from achieving marginal coverage when high coverage on one subset compensates for poor coverage on the other.

## 4 Conformal Mixed-Integer Constraint Learning

In this section, we introduce our proposed *Conformal Mixed-Integer Constraint Learning* (C-MICL) framework, integrating conformal prediction into MICL problems to provide probabilistic guarantees on the ground-truth feasibility of feasible solutions. The core idea of C-MICL is to conformalize the learned constraint $y = \hat{h}(x) \in \mathcal{Y}$, thereby constructing a statistically valid uncertainty set that the predictions must satisfy. The general C-MICL problem can be written as follows:

$$\begin{aligned}
\min_{x,z} \quad & f(x, z) \\
\text{s.t.} \quad & g(x, z) \le 0 \\
& (x, z) \in \mathcal{X} \\
& \mathcal{C}_\alpha(x) \subseteq \mathcal{Y}
\end{aligned} \tag{C-MICL}$$

where $\mathcal{C}_\alpha(x)$ is a conformal set for the decision variable $x$ based on the learned model $\hat{h}(x)$, a target coverage level $\alpha$ and a calibration set $\mathcal{D}_{\text{cal}}$. In the following, we detail how to construct $\mathcal{C}_\alpha(x)$ for both regression and classification settings, and how the constraint $\mathcal{C}_\alpha(x) \subseteq \mathcal{Y}$ can be formulated using MIP. Importantly, our approach remains compatible with the existing definition of ground-truth feasibility provided in Definition 3.1, $h(x) \in \mathcal{Y}$, requiring no modifications to the underlying feasibility notion.

**Remark 4.1.** *The feasible regions defined by the oracle constraint* $h(x) \in \mathcal{Y}$ *and the conformal set containment* $\mathcal{C}_\alpha(x) \subseteq \mathcal{Y}$ *do not exhibit a clear inclusion relationship in general. In particular, neither region necessarily contains the other, and therefore one is not more conservative nor yields larger/smaller optimal values in general.*

We assume that the model $\hat{h}(x)$ is trained on $\mathcal{D}_{\text{train}}$ and considered fixed thereafter. The conformal sets are then constructed using a disjoint calibration dataset $\mathcal{D}_{\text{cal}}$ of size $N$. This approach preserves the theoretical coverage guarantees from Theorem 3.1 and Lemma 3.1 without requiring model retraining [57]. To establish formal guarantees for the C-MICL approach, we introduce the following assumption about the conformal coverage of the feasible region of the C-MICL problem.

**Assumption 4.1.** *(Conditional Independence of Feasibility and Coverage): The events of C-MICL feasibility and conformal coverage are conditionally independent given ground-truth feasibility.*

This assumption states that the C-MICL constraints do not systematically bias the coverage properties of conformal prediction sets, conditional on the *true* (unknown) feasibility status of $h(x)$. That is, we assume that whether a point is C-MICL feasible, i.e., $(x', z') \in \mathcal{F}_N = \{(x, z) \in \mathcal{X} : g(x, z) \leq 0, \mathcal{C}_\alpha(x) \subseteq \mathcal{Y}\}$ does not affect the probability that its conformal set contains the true function value, $h(x') \in \mathcal{C}_\alpha(x')$, given its ground-truth feasibility ($h(x') \in \mathcal{Y}$) or infeasibility ($h(x') \notin \mathcal{Y}$). We provide a detailed discussion and interpretation of this assumption in Appendix B.3.

Under this condition and ground-truth feasibility conformal coverage guarantee (Lemma 3.1), we establish our main theoretical result in the following theorem, which we prove in Appendix B.2. This result provides a probabilistic certificate for the ground-truth feasibility of feasible solutions to the C-MICL problem.

**Theorem 4.1.** *Let $\mathcal{F}_N = \{(x, z) \in \mathcal{X} : g(x, z) \leq 0, \mathcal{C}_\alpha(x) \subseteq \mathcal{Y}\}$ be the feasible region of the C-MICL problem. Under the same conditions of Lemma 3.1 and Assumption 4.1, if $\mathcal{F}_N$ is nonempty, then any feasible solution $(x', z') \in \mathcal{F}_N$ is ground-truth feasible with probability at least $1 - \alpha$:*

$$\mathbb{P}\left(h(x') \in \mathcal{Y} \mid (x', z') \in \mathcal{F}_N\right) \geq 1 - \alpha. \tag{3}$$

## 4.1 Regression Conformal MICL

For the regression setting, we adopt a two-step procedure to construct a predictive model $\hat{h}(x)$ and then quantify its uncertainty through an auxiliary uncertainty function $\hat{u}(x)$. Specifically, in the first step we use $\mathcal{D}_{\text{train}}$ to train a predictive regression model $\hat{h}(x)$. Then, we use the set of absolute residuals of the model in the training set $\mathcal{D}_{\text{train}}^{\text{res}} = \{(x_i, |\hat{h}(x_i) - y_i|)\}_{i=1}^{N_{\text{train}}}$ to train the secondary regression model $\hat{u}(x)$, which estimates the prediction uncertainty of $\hat{h}(x)$ as a function of the input variable $x$ [61]. While alternative one-step approaches exist for simultaneously estimating both prediction and uncertainty models (often by leveraging model-specific choices of $\hat{h}(x)$), we adopt this two-step procedure because it is simple, broadly applicable, and model-agnostic. For a discussion of joint training techniques, we refer the reader to Angelopoulos et al. [57]. Note that our framework imposes no restrictions on the choice of predictive model for $\hat{u}(x)$; however, to ensure integration with the optimization formulation, we assume that $\hat{u}(x)$ admits a MIP representation.

Although $\hat{u}(x)$ provides a heuristic estimate of uncertainty, it can still suffer from misspecification due to data noise, bias, or limited training. To address this and provide formal guarantees on predictive reliability, we apply conformal prediction to calibrate the uncertainty estimates. Specifically, for each point in the calibration dataset $\mathcal{D}_{\text{cal}}$, a conformal score is computed as $s(x_i, y_i) = \frac{|\hat{h}(x_i) - y_i|}{\hat{u}(x_i)}$, following the procedure in Lei et al. [62]. Then, as outlined in Section 3.2, the conformal quantile $\hat{q}_{1-\alpha} = \text{Quantile}\left(s(x_1, y_1), \ldots, s(x_N, y_N); (1-\alpha)(1 + 1/N)\right)$ is computed based on the set of conformal scores. Using this quantile the conformal set constraint can be expressed as:

$$\mathcal{C}_\alpha(x) \subseteq \mathcal{Y} \iff [\hat{h}(x) \pm \hat{q}_{1-\alpha} \cdot \hat{u}(x)] \subseteq [\underline{y}, \overline{y}] \tag{4}$$

This set is defined by a pair of algebraic expressions that further constrain the lower and upper bounds on $y$ and can be directly embedded into the optimization formulation (see Appendix C). In summary, C-MICL in the regression setting involves (i) training an uncertainty model $\hat{u}(x)$, (ii) computing the conformal quantile offline (so that the formulation size remains independent of $N$), and (iii) expressing the valid region for $y$ through MIP constraints in terms of $\hat{h}(x)$ and $\hat{u}(x)$.

## 4.2 Classification Conformal MICL

For the classification setting, C-MICL does not require integrating an additional uncertainty model into the formulation. Instead, score functions can be directly computed from the predictor $\hat{h}(x) = \hat{\mathbb{P}}(Y \mid X = x)$, which typically involves a nonlinear transformation such as softmax [55]. However, in MICL, it is common practice to remove the final nonlinear transformation and operate directly on the pre-activation values of the last layer, commonly referred to as logits, to preserve linearity [8]. This approach does not affect the classification result: since softmax and sigmoid functions are monotonic, the class with the highest logit still corresponds to the highest probability.

To enable integration into MIP, we propose a valid conformal score function that operates directly in logit space. Specifically, we define the conformal score as:

$$s\left(x_i, y_i\right) = -\sum_{k \in \mathcal{K}} y_i^k \cdot \hat{h}\left(x_i\right)^k \tag{5}$$

where we select the negative of the logit output corresponding to the correct class, assuming that $y_i$ is one-hot encoded. This formulation preserves the validity of the conformal method while enabling exact and efficient representation within a linear optimization model.

Given $\hat{q}_{1-\alpha} = \text{Quantile}\left(s(x_1, y_1), \ldots, s(x_N, y_N); (1 - \alpha)(1 + 1/N)\right)$ as defined in Section 3.2, we define the conformal sets in the classification setting as:

$$\mathcal{C}_\alpha(x) \subseteq \mathcal{Y} \iff -\hat{h}(x)^k > \hat{q}_{1-\alpha} \quad \forall\, k \in \mathcal{K}^{\text{und}} \tag{6}$$

which ensures that only desired classes $k \in \mathcal{K}^{\text{des}}$ are included in the conformal prediction set, by retaining those whose conformal scores lie below the quantile threshold. This behavior can be captured within a MIP framework by introducing an indicator constraint of the form $\mathbb{1}\{-\hat{h}(x)^k \leq \hat{q}_{1-\alpha}\}$ for all $k \in \mathcal{K}$, and enforcing it to be active only for desired classes $k \in \mathcal{K}^{\text{des}}$. This logical constraint can be further reformulated as a set of algebraic mixed-integer inequalities using standard big-M techniques; we refer the reader to Appendix C for further details on this reformulation.

**Remark 4.2.** *The functional forms chosen for the score functions $s(x)$ and feasibility sets $\mathcal{Y}$ serve primarily to illustrate the algorithmic implementation of C-MICL across broad classes of regression and classification problems within optimization formulations. However, any alternative score functions (see [57]) and feasibility sets can be used, provided they admit MIP representations [41].*

## 5    Computational Experiments

We empirically validate our Conformal MICL (C-MICL) approach in both regression and multi-class classification settings. In each case, we benchmark against standard single-model MICL methods as well as the ensemble-based heuristic W-MICL proposed by Maragno et al. [7].[3] We denote the latter as W-MICL($P$), where $P$ refers to the ensemble size. For both settings, we evaluate performance on 100 randomly generated optimization instances, each defined by a sampled cost vector. This approach considers the calibration set $\mathcal{D}_{\text{cal}}$ as fixed and allows us to sample from the feasible region for each C-MICL formulation. This allows us to empirically validate the theoretical guarantee in Theorem 4.1, suggesting ground-truth feasibility of optimal solutions that are feasible by construction. Moreover, in the Appendix E we empirically verify that the assumptions from Theorem 4.1 approximately holds in our experimental settings for both regression and classification problems.

In the following, we report: (i) the empirical ground-truth feasibility rate, (ii) the relative distance of optimal value $\Delta\%_i = \frac{f_i^* - f_{\text{C-MICL}}^*}{f_{\text{C-MICL}}^*} \cdot 100\%$ for each baseline method $i$ when compared to ours, and (iii) the average CPU time required to solve each instance. The results shown in the main text correspond to a target coverage level of $\alpha = 10\%$ across 100 runs with 95% confidence intervals (CIs) detailed in Appendix E. Additional results for $\alpha = 5\%$ are available in Appendix E.1 and E.2. Key architectural and hyperparameter choices are summarized in the main text, while full details on hyperparameter selection, training, and optimization formulations are provided in Appendix D.

We focus our case studies on mixed-integer linear programming (MILP) examples, which are faster to solve in practice. This choice is made without loss of generality, as all proposed formulations and theoretical guarantees remain valid for both mixed-integer linear and nonlinear programs. All optimization problems were solved using the MILP solver Gurobi v12.0.1 with a relative optimality gap of 1% [63]. Machine learning models were implemented using scikit-learn and PyTorch, and subsequently integrated into Pyomo-based [64] optimization formulations via the open-source library OMLT [65]. The code required to reproduce all numerical experiments presented in this section is publicly available in our GitHub repository,[4] which additionally includes detailed tutorials on conformal prediction and its integration into Pyomo-based optimization formulations.

---

[3]Specifically, we implement the formulation proposed in Section 3.1 of Maragno et al. [7] using a bootstrap sample proportion of 0.5 as done by the authors.

[4]GitHub: https://github.com/dovallev/c-micl

## 5.1 Regression setting

We consider the optimal design and operation of a membrane reactor for methane aromatization and hydrogen production following Carrasco and Lima [66]. The goal is to determine five input variables (reactant flows, operating temperature, and reactor dimensions) to minimize operational cost while ensuring a product flow $\geq 50$, defining the feasible region $\mathcal{Y}$. The true system behavior is modeled by a set of ordinary differential equations serving as the oracle $h(X)$. In this setting, $y \in [0, 100]$ defines an underlying regression problem. To generate training data, we sample 1,000 input points from $\mathcal{X}$ and evaluate $y$ via the discretized ODEs with added Gaussian noise to simulate measurement uncertainty [67]. See Appendix D for additional details.

We train baseline MICL models using the full dataset, following established methodologies for different predictor types: ReLU NNs [9], GBTs [11], RFs [16], and LMDTs [17]. To evaluate the performance of the robust ensemble heuristic W-MICL, we follow the approach of Maragno et al. [7], training ensembles of size $P \in \{5, 10, 25, 50\}$ using the same cross-validated hyperparameters as their corresponding single models as described in Appendix D.1. Finally, we implement our proposed C-MICL method, which only requires training two models: $\hat{h}(x)$ and $\hat{u}(x)$. We use the same model architectures and hyperparameters as the baselines, allocating 80% of the data for training, and the remaining 20% for conformal calibration. All base models shared a common architecture for the uncertainty model: a ReLU NN with two hidden layers, each with 32 units.

Figure 1 presents the main results, comparing the ground truth feasibility coverage of the various methods across these instances for $\alpha = 10\%$. The figure shows that, across all underlying models, our proposed C-MICL method consistently achieves the target ground-truth feasibility rate, as guaranteed by Theorem 4.1. In sharp contrast, none of the baseline MICL methods reach the desired coverage level, regardless of the machine learning model employed. Moreover, the W-MICL heuristic from Maragno et al. [7] only empirically approaches the target coverage when using neural networks. These results highlight the importance of theoretical guarantees: in settings where no reliable machine learning model is available to support MICL or W-MICL approaches, our Conformal MICL framework ensures ground-truth feasibility by construction. Furthermore, the level of probabilistic guarantee can be flexibly controlled by the parameter $\alpha$, allowing practitioners to tune the method to any desired confidence level in $(0, 1)$ (with $\alpha = 0.05$ included in Appendix E). In the same appendix, violin plots of the true constraint values $h(x)$ for both $\alpha$ levels illustrate the magnitude of violations, showing that C-MICL not only incurs the fewest violations but also achieves the smallest violation magnitudes when they occur.

Figure 2 presents the relative distance in optimal objective value between each method and our proposed C-MICL approach, while Figure 3 reports the average CPU time (in seconds) required to solve the 100 optimization instances. In both figures, lighter-colored bars denote methods that failed to meet the desired feasibility rate shown in Figure 1. Results correspond to ReLU NNs, which exhibited the strongest empirical performance in terms of feasibility.

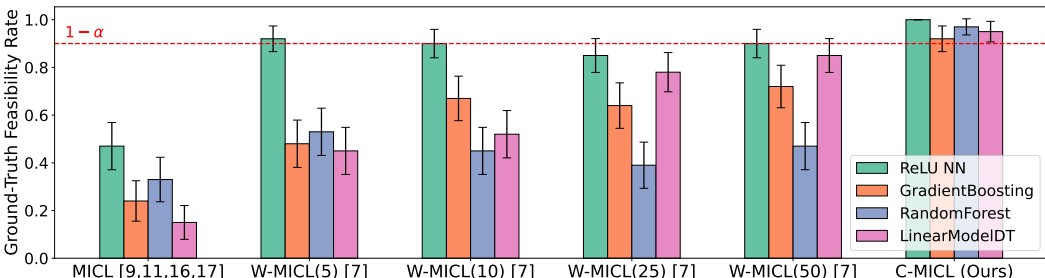

Figure 1: Empirical ground-truth feasibility rate of optimal solution across MICL methods on 100 optimization problem instances. Our Conformal MICL approach (rightmost bars) consistently achieves the target ground-truth feasibility rate $\geq 90\%$ *regardless of the underlying base model used*. Previous methods demonstrate variable performance below this theoretical guarantee. Error bars correspond to 95% confidence intervals for ground-truth feasibility rates across 100 runs.

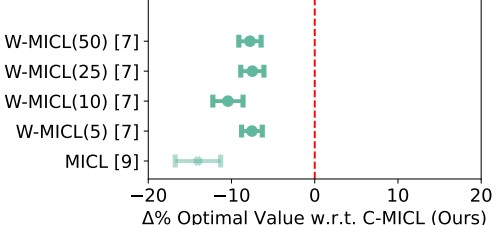 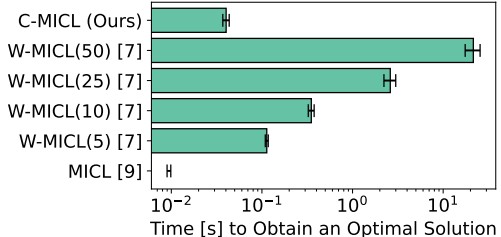

Figure 2: Percentage difference in optimal value from NN-based MICL methods compared to C-MICL. Results show an average 7% reduction in objective value compared to heuristics, indicating a modest trade-off for feasibility guarantees. MICL (lighter interval) does not achieve the desired ground-truth feasibility rate. We present 95% confidence intervals across 100 runs.

Figure 3: Average computational time to find optimal solution from NN-based MICL methods. Our C-MICL approach (top) is two orders of magnitude faster than previous methods. MICL (lighter bar) does not achieve the desired empirical feasibility rate. Error bars represent 95% confidence intervals across 100 runs.

Figure 2 shows that our proposed C-MICL method achieves the desired feasibility guarantees with only a modest average reduction of 7% in objective value compared to heuristic approaches. This small trade-off highlights that C-MICL remains highly competitive while offering formal probabilistic guarantees on feasibility. Figure 3 further highlights the computational advantages of our approach, showing that it achieves optimal solutions several orders of magnitude faster than the baseline methods. In summary, our method offers a compelling combination of computational speedup and theoretical probabilistic guarantees, while preserving comparable objective performance.

## 5.2   Multi-classification setting

We consider a food basket design problem aimed at minimizing the cost of a basket composed of 25 commodities, subject to nutritional constraints covering 12 nutrients and a learned palatability requirement [68]. Palatability, reflecting the appeal or acceptability of the basket to recipients, is inferred from historical data on previously deployed baskets and community feedback. Following the formulation in Fajemisin et al. [1], we impose a minimum palatability threshold of 0.5 (consistent with the value proposed in Maragno et al. [7]) to ensure that the basket is not only nutritionally adequate and cost-effective, but also socially and culturally appropriate.

We use the dataset of 5,000 food baskets from Maragno et al. [7], where palatability scores range continuously in $[0, 1]$. To frame this as a classification task, we discretize the scores into four categories (*bad*, *regular*, *good*, and *very good*), using thresholds at 0.25, 0.5, and 0.75, respectively. Consistent with Maragno et al. [7], we constrain the optimization problem to only include baskets labeled *good* or *very good*. As no mechanistic oracle exists for palatability, we train a regression neural network on all 5,000 samples to approximate this function. The model's predictions are thresholded to produce the categorical ground truth used in our experiments. For a consistent comparison across MICL, W-MICL, and C-MICL, we benchmark all methods on a subset of 2,500 baskets data points.

We evaluate classification performance using ReLU NNs [9], the only model type readily supported by OMLT for classification tasks [65]. Despite this, our method remains model-agnostic, as demonstrated in the regression case study. For MICL and W-MICL (with ensemble sizes $P = \{5, 10\}$), we train on all 2,500 data points. For our proposed C-MICL approach, which requires only one trained model, we use 2,300 points for training and reserved 200 for conformal calibration. Full details on network architecture and training hyperparameters are provided in Appendix D.2.

As shown in Figure 4, only our C-MICL method achieves the desired empirical coverage at $\alpha = 10\%$, in line with Theorem 4.1. This highlights the importance of formal statistical guarantees, especially under data constraints and high-dimensional inputs. Figure 5 shows the relative optimality gaps across methods, with differences averaging around 1%. However, none of the baselines yield implementable solutions, as only our method satisfies the desired empirical coverage guarantee. Figure 6 shows that C-MICL achieves solution times comparable to solving a single MICL model and is significantly faster than the W-MICL heuristics. In both figures, lighter bars indicate methods that do not meet

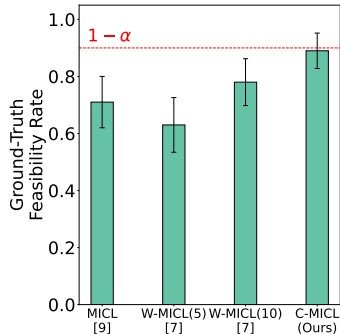 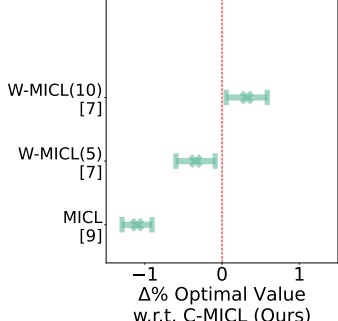 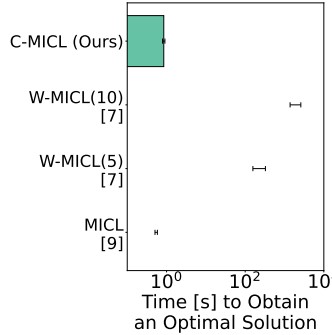

Figure 4: Empirical ground-truth feasibility rate of optimal solution across MICL methods. Our C-MICL approach (right-most bar) is the *only* method that consistently satisfies the target ground-truth feasibility rate $\geq 90\%$ across 100 runs. Error bars represent 95% CIs.

Figure 5: Percentage difference in optimal value from NN-based MICL methods compared to C-MICL. Difference is around 1% however all baseline approaches (lighter intervals) fail to achieve the desired feasibility guarantee. Results across 100 runs, with 95% CIs.

Figure 6: Average computational time to find optimal solution from NN-based MICL methods, across 100 runs. C-MICL achieves solution times comparable to single-model MICL, and is significantly faster than the W-MICL heuristics. Error bars represent 95% CIs.

the target empirical coverage. In summary, our method is the only one to guarantee ground-truth feasibility, preserves objective performance, and scales efficiently in classification tasks.

## 6  Discussion

Our work introduces C-MICL, a model-agnostic framework for incorporating learned constraints into optimization problems with probabilistic feasibility guarantees. C-MICL supports both regression and classification tasks, leverages conformal prediction to construct uncertainty sets, and encodes these into MIP-compatible formulations. It avoids the scalability bottlenecks of ensemble-based methods by requiring training at most two models, achieving substantial computational gains while maintaining comparable high-quality solutions. Furthermore, since our proposed method separates training from optimization, the size of the resulting optimization formulation is independent of the amount of data used to train the models, unlike previous end-to-end approaches (e.g., Zhao et al. [48]). Similarly, C-MICL can be integrated with existing methodologies aimed at improving the solution quality of standard MICL and W-MICL formulations, such as the enlarged convex hull trust-region approach proposed in Maragno et al. [7] or the trust-region-filter framework in Biegler [18]. C-MICL extends naturally to learning objective functions, providing probabilistic guarantees on true objective values, or to multiple learned constraints, representing a promising direction for future research on conformal-enhanced optimization.

One limitation of our work, common to general conformal prediction methods, is that when the underlying machine learning model performs poorly, the resulting conformal sets can become too wide and uninformative, potentially rendering the C-MICL problem infeasible. Moreover, although we believe the conditional independence assumption between feasibility and coverage is reasonable in our setting, it may fail in cases where the feasible region is biased toward data regimes in which conformal coverage does not hold. Nevertheless, an assumption of this nature is necessary to handle the inherent dependence between C-MICL feasible solutions and the calibration set $\mathcal{D}_{\text{cal}}$, which directly defines the feasible region and would otherwise invalidate standard conformal guarantees. This represents a theoretical compromise for integrating conformal prediction into constrained optimization and enables statistical guarantees that were previously unattainable in MICL approaches.

This work contributes to the integration of machine learning and mathematical optimization by providing a principled framework for embedding data-driven constraints into decision-making processes with statistical guarantees. By ensuring marginal feasibility without access to the true constraint function, C-MICL advances the state of constraint learning, extends predict-then-optimize pipelines, and offers a tractable approach to optimizing over uncertain, learned constraints in real-world applications.

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

# A  Machine Learning Models as MIPs

In the following we provide an overview on how to formulate off-the-shelf *pre-trained* machine learning models as mixed-integer programs (MIP) into constraint learning formulations.

## A.1  ReLU Neural Networks

We consider feed-forward neural networks composed of fully connected layers with ReLU activations, which can be exactly represented using MIP [9, 10]. Let $x$ denote the input variables to the network and $y$ its output variables. All other internal quantities (preactivations, activations, and indicator variables) are distinct auxiliary decision variables introduced solely for encoding the network structure. Therefore, each neuron computes a transformation of the form:

$$a = \max\left(0, w^\top s + b\right), \tag{7}$$

where $s$ are the activations from the previous layer, and $(w, b)$ are fixed, trained parameters. The ReLU nonlinearity is encoded via the so called *big-M formulation*:

$$\begin{aligned}
a &\geq w^\top s + b \\
a &\leq w^\top s + b - (1 - \delta)L \\
a &\leq \delta U \\
\delta &\in \{0, 1\}
\end{aligned} \tag{8}$$

where $a$ is the activation output, $\delta$ is a binary indicator for whether the neuron is active, and $L, U$ are tight lower and upper bounds on the pre-activation $w^\top s + b$. The full network is encoded layer by layer starting from the inputs $x$, recursively defining internal pre-activations and activations, until reaching the output layer $y$.

For further details and alternatives on MIP reformulations of ReLU neural networks, we refer readers to a recent review by Huchette et al. [69]. While multiple equivalent encodings exist, they yield the same global solution [53], differing only in computational efficiency, and thus do not affect our discussion on ground-truth feasibility.

## A.2  Tree Ensembles and Linear-Model Decision Trees

We encode trained tree ensembles (including random forests and gradient-boosted trees) as well as linear-model decision trees using a unified MIP formulation based on leaf selection [11, 16, 17]. Let $\mathcal{T}$ denote the set of decision trees in the ensemble, and $\mathcal{L}_t$ the set of leaf nodes in tree $t \in \mathcal{T}$. Each tree partitions the input space into disjoint regions (leaves), each associated with a constant prediction value. We introduce binary variables $r_{t,\ell} \in \{0, 1\}$ indicating whether leaf $\ell \in \mathcal{L}_t$ of tree $t$ is selected. A valid configuration activates exactly one leaf per tree:

$$\sum_{\ell \in \mathcal{L}_t} r_{t,\ell} = 1 \quad \forall t \in \mathcal{T} \tag{9}$$

Each internal node in a tree performs a threshold split of the form $x_i < v_{i,j}$, where $v_{i,j}$ is a threshold value for input feature $x_i$. We introduce binary variables $w_{i,j} \in \{0, 1\}$ to model these comparisons, where $w_{i,j} = 1$ if, and only if, the condition is satisfied. Monotonicity constraints $w_{i,j} \leq w_{i,j+1}$ ensure consistency across thresholds, reflecting the ordered structure of decision splits.

Each leaf $\ell$ in tree $t$ is reachable only if all the splitting conditions on the path to that leaf are satisfied. For each split node $s$ in tree $t$, let $i(s)$ be the split variable and $j(s)$ the corresponding threshold index. Let Left$_{t,s}$ and Right$_{t,s}$ denote the sets of leaf nodes in the left and right subtrees of $s$. We enforce:

$$\sum_{\ell \in \text{Left}_{t,s}} r_{t,\ell} \leq w_{i(s),j(s)}, \qquad \sum_{\ell \in \text{Right}_{t,s}} r_{t,\ell} \leq 1 - w_{i(s),j(s)} \quad \forall t \in \mathcal{T}, s \in \text{Splits}(t) \tag{10}$$

The final prediction is computed as a weighted sum of the selected leaves: the weights are stage-dependent for gradient-boosted trees [11], uniform across trees for random forests [16], or equal to a linear function of the input features within the selected leaf for linear-model decision trees [17].

# B  Proof of Main Results

## B.1  Mondrian Conformal Prediction

Given calibration data $\{(X_1, Y_1), \ldots, (X_N, Y_N)\}$, Mondrian conformal prediction is a general framework that extends full conformal prediction by enforcing *conditional coverage guarantees* across a pre-defined set of *finite* groups defined in terms of both the features variables $X$ and the outcome variable $Y$ [57, 60]. Formally, for a group-indicator function $g(X, Y) \in \{1, \ldots, K\}$, Mondrian conformal prediction construct conformal sets $\mathcal{C}_\alpha(X_{N+1})$ such that

$$\mathbb{P}(Y_{n+1} \in \mathcal{C}_\alpha(X_{N+1}) \mid g(X_{N+1}, Y_{N+1}) = k) \geq 1 - \alpha,$$

for all groups $k \in [K] = \{1, \ldots K\}$.

The core insight of Mondrian conformal prediction lies in computing different conformal quantiles for each group $k \in [K]$. Then, for a test point $X_{n+1}$, we generate *hypothetical* points $(X_{N+1}, y)$ for each possible value of the outcome variable $y$, and compare the conformal score $s(X_{N+1}, y)$ to the conformal scores $\{s(X_i, Y_i)\}_{i \in \mathcal{I}_{g(X_{N+1}, y)}}$ for all calibration data points *in the same group as* $(X_{N+1}, y)$. That is, the group-specific comparison is performed for $i \in \mathcal{I}_{g(X_{N+1}, y)} = \{(i \in [N] : g(X_i, Y_i) = g(X_{N+1}, y)\}$.

This approach yields the following *Mondrian conformal sets*

$$\mathcal{C}_\alpha(X_{N+1}) = \{y : s(X_{N+1}, y) \leq \hat{q}_{1-\alpha}^y\}, \tag{11}$$

where $\hat{q}_{1-\alpha}^y$ is the group-specific quantile

$$\hat{q}_{1-\alpha}^y = \text{Quantile}\left(\{s(X_i, Y_i)\}_{i \in \mathcal{I}_{g(X_{N+1}, y)}}; (1-\alpha)(1 + 1/|\mathcal{I}_{g(X_{N+1}, y)}|)\right).$$

The conditional coverage guarantee is formalized in the following theorem:

**Theorem B.1** (Theorem 4.11 in [57]). *Assume that $(X_1, Y_1), \ldots, (X_{N+1}, Y_{N+1})$ are exchangeable random variables and $s$ is a symmetric score function. Then, the Mondrian conformal set $\mathcal{C}_\alpha(X_{N+1})$ from eq. (11) satisfies the conditional coverage guarantee*

$$\mathbb{P}(Y_{n+1} \in \mathcal{C}_\alpha(X_{N+1}) \mid g(X_{N+1}, Y_{N+1}) = k) \geq 1 - \alpha \tag{12}$$

*for all groups $k \in \{1, \ldots K\}$ with $\mathbb{P}(g(X_{N+1}, Y_{N+1}) = k) > 0$, for any user-specified coverage level $\alpha \in (0, 1)$.*

Thus, Lemma 3.1 directly applies Theorem B.1 using the group identifier function $g(x, y) = \mathbb{1}(y \in \mathcal{Y})$, which separates the data according to the ground-truth feasibility condition. Note that the condition $\mathbb{P}(g(X_{N+1}, Y_{N+1}) = k) > 0$ from Theorem B.1 assumes that both feasible and infeasible points can be sampled from the underlying distribution $\mathcal{P}_{XY}$, which is given by construction of the constraint learning optimization problem.

## B.2  Proof of Theorem 4.1

*Proof.* We first show that Lemma 3.1 and Assumption 4.1 imply that we have appropriate coverage for the feasible points of the C-MICL problem $\mathcal{F}_N = \{(x, z) \in \mathcal{X} : g(x, z) \leq 0, \mathcal{C}_\alpha(x) \subseteq \mathcal{Y}\}$.

By the law of total probability,

$$
\begin{aligned}
\mathbb{P}(h(x) \in \mathcal{C}_\alpha(x) \mid (x, z) \in \mathcal{F}_N) =& \mathbb{P}(h(x) \in \mathcal{C}_\alpha(x) \mid (x, z) \in \mathcal{F}_N, h(x) \in \mathcal{Y})\mathbb{P}(h(x) \in \mathcal{Y} \mid (x, z) \in \mathcal{F}_N) \\
&+ \mathbb{P}(h(x) \in \mathcal{C}_\alpha(x) \mid (x, z) \in \mathcal{F}_N, h(x) \notin \mathcal{Y})\mathbb{P}(h(x) \notin \mathcal{Y} \mid (x, z) \in \mathcal{F}_N) \\
=& \mathbb{P}(h(x) \in \mathcal{C}_\alpha(x) \mid h(x) \in \mathcal{Y})\mathbb{P}(h(x) \in \mathcal{Y} \mid (x, z) \in \mathcal{F}_N) \\
&+ \mathbb{P}(h(x) \in \mathcal{C}_\alpha(x) \mid h(x) \notin \mathcal{Y})\mathbb{P}(h(x) \notin \mathcal{Y} \mid (x, z) \in \mathcal{F}_N) \\
\geq& (1 - \alpha)\mathbb{P}(h(x) \in \mathcal{Y} \mid (x, z) \in \mathcal{F}_N) + (1 - \alpha)\mathbb{P}(h(x) \notin \mathcal{Y} \mid (x, z) \in \mathcal{F}_N) \\
=& (1 - \alpha)[\mathbb{P}(h(x) \in \mathcal{Y} \mid (x, z) \in \mathcal{F}_N) + \mathbb{P}(h(x) \notin \mathcal{Y} \mid (x, z) \in \mathcal{F}_N)] \\
=& (1 - \alpha),
\end{aligned}
$$

where the second equality follows from the conditional independence assumption 4.1

$$\mathbb{P}(h(x) \in \mathcal{C}_\alpha(x) \mid (x, z) \in \mathcal{F}_N, h(x) \in \mathcal{Y}) = \mathbb{P}(h(x) \in \mathcal{C}_\alpha(x) \mid h(x) \in \mathcal{Y}),$$
$$\mathbb{P}(h(x) \in \mathcal{C}_\alpha(x) \mid (x, z) \in \mathcal{F}_N, h(x) \notin \mathcal{Y}) = \mathbb{P}(h(x) \in \mathcal{C}_\alpha(x) \mid h(x) \notin \mathcal{Y}).$$

Moreover, the first inequality follows from the ground-truth feasibility conformal coverage guarantee in Lemma 3.1 at level $\alpha$

$$\min\left(\mathbb{P}(h(x) \in \mathcal{C}_\alpha(x) \mid h(x) \in \mathcal{Y}), \mathbb{P}(h(x) \in \mathcal{C}_\alpha(x) \mid h(x) \notin \mathcal{Y})\right) \geq 1 - \alpha,$$

where the probabilities are taken with respect to both the calibration data $\mathcal{D}_{\text{cal}}$ and the test point.

The previous result says that when a solution $(x', z') \in \mathcal{F}_N$ is feasible, its true function value $h(x')$ falls within $\mathcal{C}_\alpha(x')$ with probability at least $1 - \alpha$. Since $\mathcal{C}_\alpha(x')$ is a subset of $\mathcal{Y}$ for feasible solutions to the C-MICL problem, whenever $h(x')$ is in $\mathcal{C}_\alpha(x')$, it must also be in $\mathcal{Y}$. Therefore,

$$\mathbb{P}(h(x') \in \mathcal{Y} \mid (x', z') \in \mathcal{F}_N) \geq \mathbb{P}(h(x') \in \mathcal{C}_\alpha(x') \mid (x', z') \in \mathcal{F}_N) \geq 1 - \alpha,$$

which proves the ground-truth feasibility guarantee.

$\square$

## B.3 Discussion on the Conditional Independence of Feasibility and Coverage Assumption

Assumption 4.1 states that, conditional on ground-truth feasibility ($h(x) \in \mathcal{Y}$ or $h(x) \notin \mathcal{Y}$), the event of C-MICL feasibility ($(x, z) \in \mathcal{F}_N = \{(x, z) \in \mathcal{X} : g(x, z) \leq 0, \mathcal{C}_\alpha(x) \subseteq \mathcal{Y}\}$) is independent of whether the conformal set contains the true function value ($h(x) \in \mathcal{C}_\alpha(x)$). To motivate Assumption 4.1 and clarify when it is plausible, we first emphasize that the ground-truth feasibility (GTF) conditional coverage guarantee from Lemma 3.1 can be achieved in a fully data-driven way (e.g., using Mondrian conformal prediction or other label-conditional conformal methods):

$$\mathbb{P}(h(x) \in \mathcal{C}_\alpha(x) \mid h(x) \in \mathcal{Y}) \geq 1 - \alpha$$
$$\mathbb{P}(h(x) \in \mathcal{C}_\alpha(x) \mid h(x) \notin \mathcal{Y}) \geq 1 - \alpha$$

However, in C-MICL we aim to guarantee coverage over the feasible region of the optimization problem $\mathcal{F}_N = \{(x, z) \in \mathcal{X} : g(x, z) \leq 0, \mathcal{C}_\alpha(x) \subseteq \mathcal{Y}\}$, i.e.,

$$\mathbb{P}(h(x) \in \mathcal{C}_\alpha(x) \mid (x, z) \in \mathcal{F}_N) \geq 1 - \alpha$$

If the predictive model $\widehat{h}(x)$ were perfect (i.e., $\widehat{h}(x) = h(x)$), then the regions $\mathcal{F}_N$ and the ground-truth feasible region $\mathcal{F}$ would coincide, and the coverage guarantee would transfer directly. However, since we are interested in the more realistic case where $\widehat{h}(x)$ is imperfect, $\mathcal{F}_N$ and $\mathcal{F}$ differ in a data-dependent way. In this case, since the feasible region $\mathcal{F}_N$ is implicitly shaped by the calibration set (via $\mathcal{C}_\alpha(x)$), there is a natural dependency between the feasible solutions of the C-MICL problem and the calibration data, which invalidates standard conformal guarantees relying on exchangeable calibration and test data. Assumption 4.1 precisely seeks to decouple this dependency: it allows us to approximate conformal coverage within $\mathcal{F}_N$ by assuming that feasibility does not systematically bias conformal validity, once conditioned on ground-truth feasibility.

To build intuition, consider partitioning $\mathcal{F}_N$ into two disjoint subsets: $\mathcal{F}_N \cap \mathcal{F}$ and $\mathcal{F}_N \cap \mathcal{F}^c$. Then, Assumption 4.1 implies that $\mathcal{F}_N \cap \mathcal{F}$ (respectively $\mathcal{F}_N \cap \mathcal{F}^c$) is not systematically biased towards a region of $\mathcal{F}$ ($\mathcal{F}^c$) that is miscalibrated. Mathematically,

$$\mathbb{P}(h(x) \in \mathcal{C}_\alpha(x) \mid (x, z) \in \mathcal{F}_N \cap \mathcal{F}) = \mathbb{P}(h(x) \in \mathcal{C}_\alpha(x) \mid (x, z) \in \mathcal{F}_N, h(x) \in \mathcal{Y})$$
$$\approx \mathbb{P}(h(x) \in \mathcal{C}_\alpha(x) \mid h(x) \in \mathcal{Y})$$
$$\mathbb{P}(h(x) \in \mathcal{C}_\alpha(x) \mid (x, z) \in \mathcal{F}_N \cap \mathcal{F}^c) = \mathbb{P}(h(x) \in \mathcal{C}_\alpha(x) \mid (x, z) \in \mathcal{F}_N, h(x) \notin \mathcal{Y})$$
$$\approx \mathbb{P}(h(x) \in \mathcal{C}_\alpha(x) \mid h(x) \notin \mathcal{Y})$$

These enable us to translate the conformal coverage guarantees from the ground-truth feasible region $\mathcal{F}$ to the feasible set $\mathcal{F}_N$ used in the optimization. Assumption 4.1 is therefore reasonable when the calibration data adequately covers the parts of the input space that intersect the feasible region $\mathcal{F}_N$, both within the ground-truth feasible region $\mathcal{F}$ and its complement $\mathcal{F}^c$. In this sense, it aligns with standard generalization assumptions that require the training and calibration data to be representative of the regions where predictions are deployed. In our experimental settings, we observe good empirical alignment between target and achieved coverage (Appendix E), suggesting that Assumption 4.1 holds reasonably well in practice in realistic data scenarios.

Alternatively, Assumption 4.1 can be approximated using more granular conditional conformal methods, by partitioning the optimization region into finer subregions and enforcing local coverage guarantees within each, which can then be translated to the feasible set $\mathcal{F}_N$. However, the assumption may break down if the feasible region $\mathcal{F}_N$ is heavily concentrated in areas where the calibration set is sparse or systematically miscalibrated.

## C    Conformal Set MIP Reformulations

The reformulation of the conformal set in the regression setting is straightforward, as the conformal prediction interval is already algebraic in nature and can be directly expressed as:

$$\mathcal{C}_\alpha(x) \subseteq \mathcal{Y} \iff \left[\hat{h}(x) \pm \hat{q}_{1-\alpha} \cdot \hat{u}(x)\right] \subseteq [\underline{y}, \overline{y}] \tag{13}$$

$$\iff \begin{cases} \hat{h}(x) + \hat{q}_{1-\alpha} \cdot \hat{u}(x) \leq \overline{y}, \\ \hat{h}(x) - \hat{q}_{1-\alpha} \cdot \hat{u}(x) \geq \underline{y} \end{cases} \tag{14}$$

Here, we include only the two non-dominated constraints i.e., those constraints not already implied by the others, to avoid redundancy. Note that these are linear constraints in terms of the estimated $\hat{h}(x)$ and $\hat{u}(x)$, which are assumed to be MIP-representable, and a constant $\hat{q}_{1-\alpha}$ computed offline during the conformalization procedure.

The classification setting introduces additional modeling complexity, as the inclusion of classes in the prediction set must be encoded using mixed-integer constraints. Specifically, the condition

$$\mathbb{1}\{-\hat{h}(x)^k \leq \hat{q}_{1-\alpha}\} \quad \forall\, k \in \mathcal{K}, \tag{15}$$

represents an indicator that is activated (i.e., equals one) if and only if class $k$ is included in the prediction set. To enforce correct classification behavior, we require that at least one desired class $k \in \mathcal{K}^{\text{des}}$ is predicted (i.e., its indicator is activated), and no undesired class $k \in \mathcal{K}^{\text{und}}$ is allowed in the prediction set. To model this, we introduce a binary decision variable for each class $k \in \mathcal{K}$:

$$w_k = \begin{cases} 1 & \text{if class } k \text{ is included in the prediction set.} \\ 0 & \text{otherwise.} \end{cases} \quad \forall\, k \in \mathcal{K} \tag{16}$$

Using these variables, the desired classification logic can be encoded via the following mixed-integer constraints:

$$\mathcal{C}_\alpha(x) \subseteq \mathcal{Y} \iff -\hat{h}(x)^k > \hat{q}_{1-\alpha} \quad \forall\, k \in \mathcal{K}^{\text{und}} \tag{17}$$

$$\iff \begin{cases} -\hat{h}(x)^k - \hat{q}_{1-\alpha} \leq M(1 - w_k) & \forall\, k \in \mathcal{K} \\ \hat{h}(x)^k + \hat{q}_{1-\alpha} + \epsilon \leq M w_k & \forall\, k \in \mathcal{K} \\ \sum_{k \in \mathcal{K}^{\text{des}}} w_k \geq 1 & \\ w_k = 0 & \forall\, k \in \mathcal{K}^{\text{und}} \end{cases} \tag{18}$$

Here, $\epsilon$ is a small numerical tolerance (e.g., $10^{-6}$) to ensure strict inequality, and $M$ is a large positive constant used in the big-M reformulation. A practical choice for $M$ is the maximum absolute value of the predicted logits observed in the calibration dataset, scaled by a safety factor (e.g., 4):

$$M = 4 \cdot \max_i \left|\hat{h}(x_i)\right| \tag{19}$$

For detailed discussion on selecting valid big-M values and their impact on computational performance, we refer the reader to Trespalacios and Grossmann [70].

Note that the conformal prediction sets $\mathcal{C}_\alpha(x)$ used in our formulation of C-MICL have predictable, well-structured forms that are always MIP-representable, making the containment constraint $\mathcal{C}_\alpha(x) \subseteq \mathcal{Y}$ tractable regardless of the underlying prediction model.

## D    Implementation Details

All computations were performed on a Linux machine running Ubuntu, equipped with eight Intel®, Xeon®, Gold 6234 CPUs (3.30 GHz) and 1 TB of RAM, utilizing a total of eight hardware threads.

### D.1    Regression

This case study focuses on the optimal design and operation of a membrane reactor system used for the direct aromatization of methane. This integrated unit enables the conversion of methane into hydrogen and benzene, achieving simultaneous chemical reaction and product separation. By selectively removing hydrogen through a membrane, the reactor leverages Le Chatelier's principle to drive the equilibrium forward, resulting in higher methane conversion rates [66].

The optimization problem involves five key decision variables: the inlet volumetric flow rate of methane ($v_0$), the inlet flow rate of sweep gas ($v_{He}$), the operating temperature ($T$), the tube diameter ($d_t$), and the reactor length ($L$). These variables were sampled uniformly within physically reasonable bounds, as shown below:

- $v_0 \sim \mathcal{U}(450, \ 1500)$ cm$^3$/h
- $v_{He} \sim \mathcal{U}(450, \ 1500)$ cm$^3$/h
- $d_t \sim \mathcal{U}(0.5, \ 2.0)$ cm
- $L \sim \mathcal{U}(10, \ 100)$ cm
- $T \sim \mathcal{U}(997.18, \ 1348.12)$ K

The resulting samples were used as initial conditions for solving the system of ordinary differential equations governing the reactor, enabling the computation of the outlet benzene flow ($F_{C_6H_6}$). The numerical integration was performed using code available in the `opyrability` repository [71]. To simulate measurement uncertainty, Gaussian noise was added to the computed benzene flows.

Table 1 shows an illustrative subset of the 1,000 data-points generated:

Table 1: Sampled decision variables and simulated benzene outlet flow.

| $v_0$ (cm$^3$/h) | $v_{He}$ (cm$^3$/h) | $T$ (K) | $d_t$ (cm) | $L$ (cm) | $F_{C_6H_6}$ (mol/h) |
|---|---|---|---|---|---|
| 1157.75 | 845.25 | 1272.93 | 0.57 | 30.67 | 37.45 |
| 773.89 | 1319.69 | 998.56 | 0.75 | 88.10 | 42.57 |
| 1484.77 | 476.68 | 1201.21 | 1.64 | 77.38 | 40.70 |
| 817.68 | 536.65 | 1123.28 | 0.72 | 86.07 | 28.26 |
| 1162.73 | 1262.03 | 1256.59 | 1.95 | 92.29 | 36.58 |

All hyperparameters used across the single-model baselines, wrapped approaches, and our proposed methods were kept consistent to ensure a fair comparison. For the the Linear-Model Decision Tree, we set the maximum depth to five, the minimum number of samples required to split an internal node to ten, and the number of bins used for discretization to forty. For the Random Forest, we used fifteen estimators, a maximum depth of five, a minimum samples split of three, and considered sixty percent of the features when looking for the best split. The Gradient Boosting Tree model was configured with fifteen estimators, a learning rate of 0.2, a maximum depth of five, a minimum of five samples per split, and sixty percent of the features considered at each split. Lastly, the ReLU Neural Network was set up with two hidden layers of 32 units each, an L2 regularization strength of 0.01, and trained for 2000 epochs using Adam optimizer.

All models were cross-validated using a 5-fold split of their respective datasets. For the single base models, the complete 1,000 point dataset was used for training and evaluation. In the case of the ensemble methods, bootstrapping the 1,000 data-point dataset was employed to generate the required 500 data-point (half sized) subsets for training following [7]. For our conformal method, a split of the training data (800 data points) was used, ensuring that the calibration data was kept separate (200 data points). The Table 2 presents the average Mean Squared Error (MSE) across all folds for each of the methods, considering that the variance of the output for the 1,000 data-points is 1.2871.

Table 2: Average MSE across folds for all Methods for reactor model.

| Predictive Method | | |
|---|---|---|
| Model | Approach | MSE |
| RandomForest | MICL | 0.1748 |
| ReLU NN | MICL | 0.0597 |
| GradientBoosting | MICL | 0.1192 |
| LinearModelDT | MICL | 0.0634 |
| RandomForest | W-MICL(5) | 0.1712 |
| ReLU NN | W-MICL(5) | 0.070 |
| GradientBoosting | W-MICL(5) | 0.1298 |
| LinearModelDT | W-MICL(5) | 0.0594 |
| RandomForest | W-MICL(10) | 0.1711 |
| ReLU NN | W-MICL(10) | 0.0699 |
| GradientBoosting | W-MICL(10) | 0.1239 |
| LinearModelDT | W-MICL(10) | 0.0568 |
| RandomForest | W-MICL(25) | 0.1696 |
| ReLU NN | W-MICL(25) | 0.0689 |
| GradientBoosting | W-MICL(25) | 0.1217 |
| LinearModelDT | W-MICL(25) | 0.0568 |
| RandomForest | W-MICL(50) | 0.1665 |
| ReLU NN | W-MICL(50) | 0.0685 |
| GradientBoosting | W-MICL(50) | 0.1194 |
| LinearModelDT | W-MICL(50) | 0.0559 |
| RandomForest | C-MICL | 0.1847 |
| ReLU NN | C-MICL | 0.0841 |
| GradientBoosting | C-MICL | 0.1402 |
| LinearModelDT | C-MICL | 0.0696 |

For all uncertainty models $\hat{u}(x)$, we employed ReLU-based Neural Networks with a shared architecture and regularization setup to maintain consistency across approaches. Specifically, each model used two hidden layers with 32 units each and an $L_2$ regularization (weight decay) coefficient of 0.001. The only hyperparameter that varied during cross-validation was the number of training epochs, which does not influence the MIP optimization outcomes but can affect the quality of the uncertainty estimates. The optimal number of training epochs identified through cross-validation were as follows: 1000 epochs for the model paired with Gradient Boosting, 3000 epochs for the one used with the ReLU Neural Network (using Adam optimizer), and 2000 epochs for both the Random Forest and Linear-Model Decision Tree variants. Table 3 displays the average MSE for the uncertainty model of each base predictor across all folds.

Table 3: Uncertainty average MSE for each base model.

| Base Model | $\hat{u}(x)$ MSE |
|---|---|
| GradientBoosting | 0.0142 |
| ReLU NN | 0.0247 |
| RandomForest | 0.0408 |
| LinearModelDT | 0.0225 |

Tables 4, 5, and 6 present the sets, parameters and decision variables of the problem, respectively.

Table 4: Sets used in the reactor optimization model.

| Symbol | Description |
|---|---|
| $\mathcal{I}$ | Set of decision variables (e.g., $\mathcal{I} = \{$v0, v_He, T, dt, L$\}$) |

Table 5: Parameters used in the reactor optimization model.

| Symbol | Description |
|--------|-------------|
| $c_i$ | Operational or design cost coefficient associated with variable $i \in \mathcal{I}$ |

Table 6: Decision variables in the reactor design model.

| Symbol | Description |
|--------|-------------|
| $x_i$ | Value of design variable $i \in \mathcal{I}$, where $\mathcal{I} = \{\texttt{v0, v\_He, T, dt, L}\}$ |
| $y$ | Predicted outlet flow of $C_6H_6$ from the surrogate model |

**Formulation.**

$$\min_{\{x_i\}_{i \in \mathcal{I}}, y} \quad \sum_{i \in \mathcal{I}} c_i x_i \tag{D.1a}$$

$$\text{s.t.} \quad 10 \leq \frac{x_L}{x_{dt}} \leq 150 \tag{D.1b}$$

$$0.75 \leq \frac{x_{v0}}{x_{v\_He}} \leq 3.0 \tag{D.1c}$$

$$20 \leq \frac{x_{v0}}{x_L} \leq 120 \tag{D.1d}$$

$$x_{v0} \leq 1.1 \cdot x_T \tag{D.1e}$$

$$\hat{h}(\boldsymbol{x}) = y \tag{D.1f}$$

$$y \geq 50 \tag{D.1g}$$

$$x_i \geq 0 \quad \forall\, i \in \mathcal{I} \tag{D.1h}$$

**Explanation of Constraints.**

- **(D.1a)**: Minimize the operating cost as a linear function of the decision variables.
- **(D.1b)**: Enforce physical bounds on the tube length-to-diameter ratio.
- **(D.1c)**: Maintain a suitable gas feed ratio between $CH_4$ and He.
- **(D.1d)**: Control the residence time via the $CH_4$ flow and reactor length.
- **(D.1e)**: Limits methane flow rate based on temperature to ensure safe and stable operation.
- **(D.1f)**: Enforces that the surrogate model output $y$, predicting $C_6H_6$ flow, is computed from the design and operation variables $\boldsymbol{x}$.
- **(D.1g)**: Requires that the predicted $C_6H_6$ outlet flow (i.e., product quality) meets or exceeds the target value of 50.
- **(D.1h)**: Enforces nonnegativity for all design variables.

### D.2 Classification

We now examine the food basket optimization problem introduced earlier and grounded in the work of Peters et al. [68]. Our analysis is based on the model developed by Fajemisin et al. [1], which aims to minimize the cost of assembling a basket consisting of 25 different commodities while satisfying nutritional requirements across 12 key nutrients. An additional constraint is placed on the palatability of the basket, which reflects how acceptable or appealing the food is to the target population. In line with the case study by Maragno et al. [7], we require a minimum palatability score of $t = 0.5$ to ensure that the resulting food baskets are not only affordable and nutritionally adequate but also culturally and socially acceptable. The dataset used in this analysis is publicly available at and published by Maragno et al. [7] here.

The palatability scores initially range continuously from 0 to 1. To transform this into a classification problem, we discretize these scores into four distinct categories: *bad*, *regular*, *good*, and *very good*. This discretization is achieved by setting thresholds at 0.25, 0.5, and 0.75. Consequently, the problem

becomes a multi-class classification task, where each food basket is assigned one of these categorical labels based on its palatability score. We restrict the optimization problem to include only food baskets that fall within the *good* and *very good* categories. This approach is inspired by the work of Maragno et al. [7], where a similar threshold of $t = 0.5$ was used, but applied in a categorical context. Table 7 displays a sample of the dataset available.

Table 7: Sample of food basket dataset with commodity quantities and palatability score. Full dataset available here.

| Beans | Bulgur | Cheese | Fish | Meat | CSB | Dates | DSM | ... | Palatability | Class |
|---|---|---|---|---|---|---|---|---|---|---|
| 0.7226 | 0 | 0 | 0 | 0 | 0 | 0 | 0.5987 | ... | 0.199 | Bad |
| 0.7860 | 0 | 0 | 0 | 0 | 0.0419 | 0 | 0.3705 | ... | 0.8049 | Very Good |
| 0.4856 | 0 | 0 | 0 | 0 | 0 | 0 | 0.2696 | ... | 0.6517 | Good |
| 0 | 0 | 0.5734 | 0 | 0 | 0 | 0 | 0.0025 | ... | 0.3220 | Regular |

In this case, we exclusively trained ReLU-based neural networks across all methods (single-model baseline, wrapped approach, and our proposed method) ensuring that all hyperparameters remained identical for a fair comparison. For the predictors, the network architecture consisted of three hidden layers with 64 units each, trained for 500 epochs with a weight decay (L2 regularization) of 0.01. Additionally, we trained an oracle model using all available data. This oracle network was configured with five hidden layers of 256 units each, trained for 1000 epochs, also with a weight decay of 0.01.

Table 8 displays the average accuracy across validation folds on the data available for each approach.

Table 8: Average accuracy across folds for all methods for the basket model.

| Predictive method | | |
|---|---|---|
| Model | Approach | Accuracy [%] |
| ReLU NN | MICL | 84.32 |
| ReLU NN | W-MICL(5) | 78.24 |
| ReLU NN | W-MICL(10) | 76.88 |
| ReLU NN | C-MICL | 83.30 |

Tables 9, 10, and 11 present the sets, parameters and decision variables of the problem, respectively.

Table 9: Sets used in the basket model.

| Symbol | Description |
|---|---|
| $\mathcal{M}$ | Set of commodities (e.g., $\mathcal{M} = \{$rice, beans, salt, sugar, ...$\}$) |
| $\mathcal{L}$ | Set of nutrients (e.g., $\mathcal{L} = \{$protein, iron, calories, ...$\}$) |
| $\mathcal{K}^{\text{des}}$ | Set of desired categories (e.g., $\mathcal{K}^{\text{des}} = \{$good, very good$\}$) |
| $\mathcal{K}^{\text{und}}$ | Set of undesired categories (e.g., $\mathcal{K}^{\text{und}} = \{$bad, regular$\}$) |

Table 10: Parameters used in the basket model.

| Symbol | Description |
|---|---|
| $\text{Nutreq}_l$ | Nutritional requirement for nutrient $l \in \mathcal{L}$ (grams/person/day) |
| $\text{Nutval}_{ml}$ | Nutrient content of commodity $m$ for nutrient $l$ (grams per gram) |
| $p_m$ | Procurement cost of commodity $m$ (in \$/metric ton) |
| $M$ | Large positive constant for big-M reformulation. |

Table 11: Decision variables in the basket model.

| Symbol | Description |
|---|---|
| $x_m$ | Quantity of commodity $m \in \mathcal{M}$ in the food basket (grams) |
| $y$ | Palatability level of the food basket |
| $w_k$ | Binary indicator for predicting desired category $k \in \mathcal{K}^{\text{des}}$ ({0,1}) |

**Formulation.**

$$\min_{\{x_m\}_{m \in \mathcal{M}}, \, y} \quad \sum_{m \in \mathcal{M}} p_m x_m \tag{D.2a}$$

$$\text{s.t.} \quad \sum_{m \in \mathcal{M}} \text{Nutval}_{ml} \cdot x_m \geq \text{Nutreq}_l, \qquad \forall \, l \in \mathcal{L} \tag{D.2b}$$

$$x_{\text{salt}} = 5 \tag{D.2c}$$

$$x_{\text{sugar}} = 20 \tag{D.2d}$$

$$\hat{h}(\boldsymbol{x}) = \boldsymbol{y} \tag{D.2e}$$

$$y_{k'} - y_k \leq M(1 - w_k) \qquad \forall \, k \in \mathcal{K}^{\text{des}}, \, k' \in \mathcal{K}^{\text{und}} \tag{D.2f}$$

$$\sum_{k \in \mathcal{K}^{\text{des}}} w_k \geq 1 \tag{D.2g}$$

$$x_m \geq 0 \qquad \forall \, m \in \mathcal{M} \tag{D.2h}$$

**Explanation of Constraints.**

- **(D.2a)**: Objective function minimizing total procurement cost of the food basket.

- **(D.2b)**: Nutritional constraints to ensure daily nutrient requirements are met.

- **(D.2c)–(D.2d)**: Fixed amounts of salt and sugar imposed (e.g., due to guidelines).

- **(D.2e)**: Palatability is computed as a function $\hat{h}(\boldsymbol{x})$ of the selected commodity quantities.

- **(D.2f)**: Big-M constraint that activates an indicator if the logit of a desired class $k \in \mathcal{K}^{\text{des}}$ is higher than the logit of all undesired classes $k' \in \mathcal{K}^{\text{und}}$, representing desired prediction. $M$ is calculated using the largest value in magnitude observed in calibration data times a enlarging safety factor (e.g., 4) as $M = 4 \cdot \max_i |\hat{h}(x_i)|$. For more information on how to calculate valid $M$ values and their impacts in optimization we refer the readers to Trespalacios and Grossmann [70].

- **(D.2g)**: Enforce at least one desired class to have larger logit that both undesired classes.

- **(D.2h)**: Ensures quantity nonnegativity.

# E  Complementary Results

To quantify uncertainty and support the statistical reliability of our results, we report 95% confidence intervals (Cls) for all key metrics: empirical feasibility rates, optimization times, relative differences in objective value, and true coverage. All uncertainty comes from 100 problem instances of optimization problems.

For feasibility rates and true coverage plots intervals are computed as the proportion of feasible solutions over meaning that we model each as a Bernoulli random variable and apply the standard error formula for proportions:

$$\text{SEM} = \sqrt{\frac{\bar{p}(1-\bar{p})}{n}},$$

where $\bar{p}$ is the sample mean and $n$ is the number of problem instances. The corresponding confidence interval is calculated using the Student's t-distribution:

$$\text{CI}_{95\%} = \bar{p} \pm t_{n-1,0.975} \cdot \text{SEM}$$

For optimization times and relative differences in objective values, we compute the sample mean, standard deviation, and standard error of the mean (SEM) for each method across instances. The 95% confidence intervals are then given by:

$$\text{CI}_{95\%} = \bar{x} \pm t_{n-1,0.975} \cdot \frac{s}{\sqrt{n}}$$

where $s$ is the sample standard deviation and $n = 100$ is the number of observations. This method assumes approximate normality of the sample means, which is reasonable due to the Central Limit Theorem given the sample size.

All error bars shown in plots correspond to these 95% confidence intervals. The factors of variability captured are due to the random generation of optimization instances and the resulting performance metrics across different methods. These intervals are reported directly in the results section and supporting figures to substantiate claims about statistical significance and performance differences.

### E.1   Regression

The following results summarize performance on the reactor case study at $\alpha = 10\%$, evaluated after solving 100 optimization instances for each method. More specifically, Figure 7 reports the average computational time required to obtain an optimal solution for each method at $\alpha = 10\%$, while Figure 8 presents the relative difference in optimal objective value between baseline methods and our proposed C-MICL at the same confidence level. Lighter bars in both figures denote methods that failed to achieve the target empirical feasibility rate, as established in Figure 1. Additionally, Figure 9 shows the distribution of true constraint values $h(x)$ for each methods for $\alpha = 10\%$. Finally, Figure 10 reports the empirical coverage over 1,000 out-of-sample data points, stratified by deciles of the true output variable $y$. For each decile, we report the proportion of instances where the true value lies within the predicted interval, demonstrating strong coverage across the output space for all baseline models. Notably, our ReLU NN-based conformal sets empirically satisfy the ground-truth feasibility coverage guarantee from Lemma 3.1, achieving coverage rates of 89.31% and 95.38% on ground-truth infeasible and feasible samples, respectively.

We report here the full results for the reactor case study at $\alpha = 5\%$, based on evaluations conducted over 100 solved optimization instances per method. Specifically, Figure 11 reports the empirical ground-truth feasibility rate achieved by each MICL approach. Figure 12 shows the average computational time needed to solve each method at $\alpha = 5\%$, and Figure 13 displays the relative difference in optimal objective value between baseline methods and our C-MICL approach. Furthermore, Figure 14 presents the distribution of true constraint values $h(x)$ for each method at $\alpha = 5\%$. Finally, Figure 15 reports the empirical coverage of the models over 1,000 out-of-sample data points, grouped by deciles of the true output $y$. Coverage is measured as the proportion of points for which the true value of $y$ falls within the predicted interval at a target level of $\alpha = 5\%$. Our ReLU NN-based conformal sets demonstrate empirical compliance with the ground-truth feasibility coverage guarantee established in Lemma 3.1, yielding coverage rates of 96.92% for ground-truth feasible instances and 96.09% for ground-truth infeasible ones.

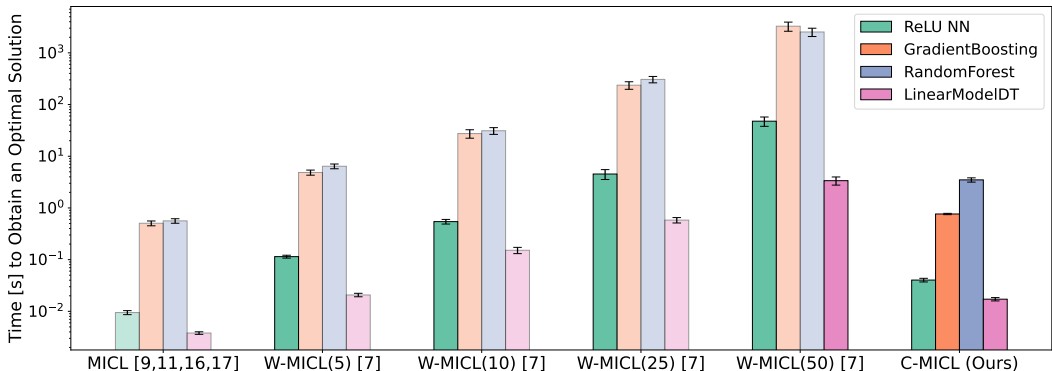

Figure 7: Average computational time to obtain an optimal solution for all methods at $\alpha = 10\%$. Our proposed C-MICL approach is comparable to single-model baselines while being orders of magnitude faster than ensemble heuristics using the same underlying base model. Lighter bars indicate methods that failed to achieve the target empirical feasibility rate.

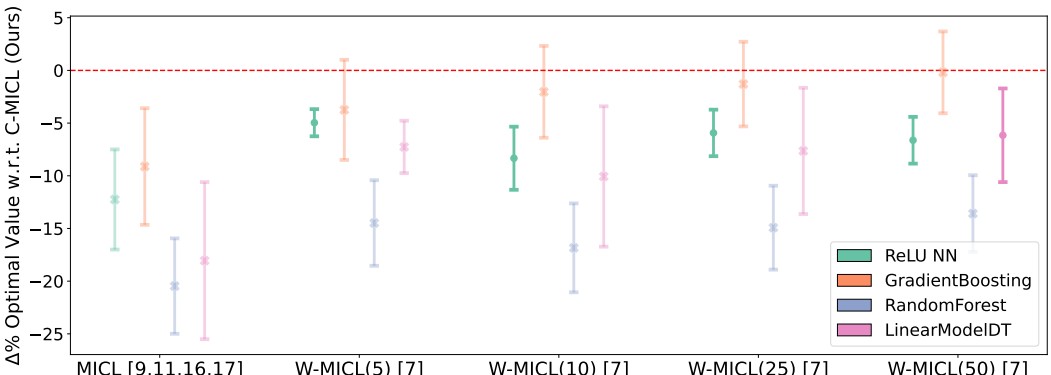

Figure 8: Relative difference in optimal objective value of baseline methods compared to our proposed C-MICL at $\alpha = 10\%$. Our approach exhibits a small difference of around 8% compared to the method that achieved empirical coverage, indicating that C-MICL attains comparable solution quality relative to valid approaches. Lighter bars marked with an "x" denote methods that failed to meet the target empirical feasibility rate.

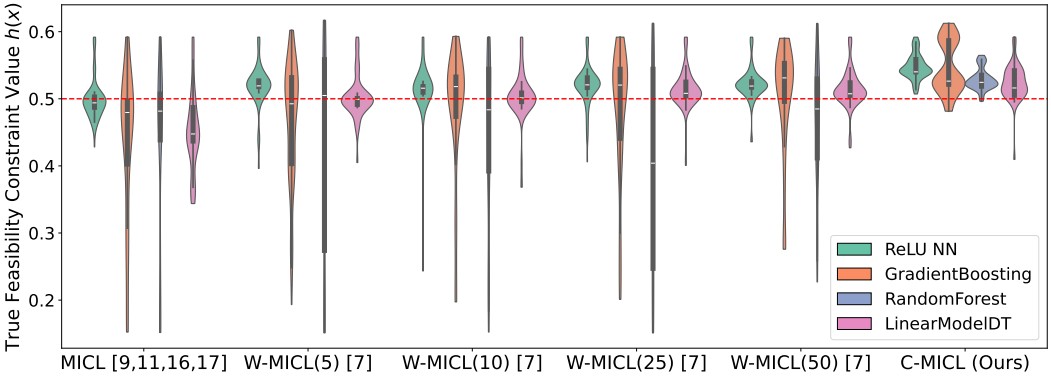

Figure 9: Distribution of true constraint values $h(x)$ for each method for $\alpha = 10\%$ The dotted line at 0.5 marks the lower bound imposed in the oracle constraint values below this line correspond to true infeasibilities. C-MICL not only yields the fewest violations but also achieves the smallest violation magnitudes when they occur.

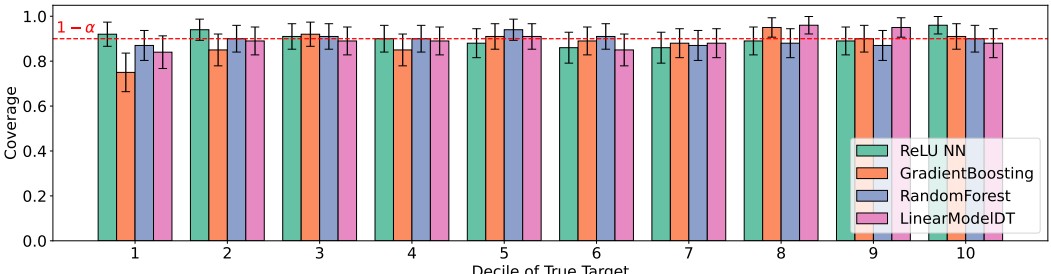

Figure 10: Empirical out-of-sample coverage across predictive models at $\alpha = 10\%$, evaluated on 1,000 test points and stratified by deciles of the true output $y$. Our conformal sets achieve strong target coverage. Crucially, the ground-truth feasibility threshold ($y \geq 50\%$) lies within the 9th and 10th deciles, where all methods attain valid empirical coverage, empirically supporting the assumptions of Theorem 4.1.

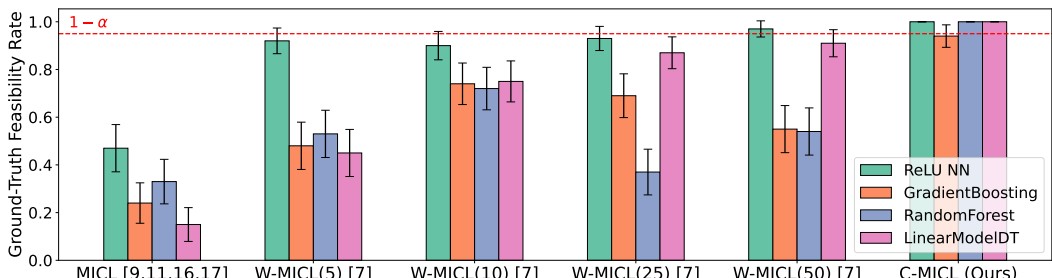

Figure 11: Empirical ground-truth feasibility rate of optimal solutions across MICL methods on 100 optimization problem instances for $\alpha = 5\%$. Our Conformal MICL approach (rightmost bars) consistently meets the target feasibility rate of $\geq 95\%$ *regardless of the underlying base model*. In contrast, existing methods show inconsistent performance and often fall short of the theoretical guarantee.

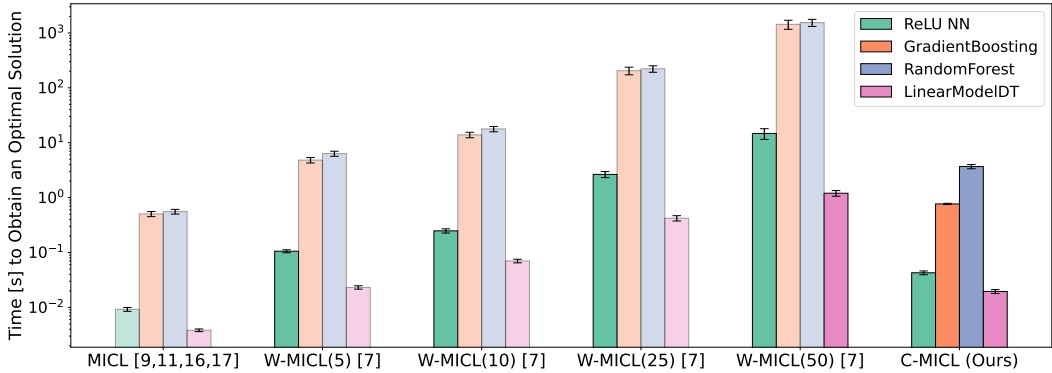

Figure 12: Average computational time to solve 100 optimization instances at $\alpha = 5\%$. Our proposed C-MICL method matches the speed of single-model baselines and is orders of magnitude faster than ensemble-based heuristics using the same base model. Lighter bars denote methods that did not reach the target empirical feasibility threshold.

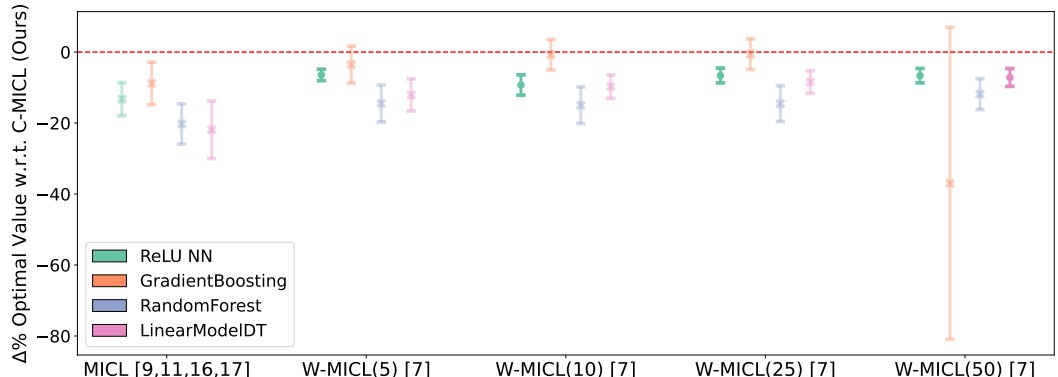

Figure 13: Relative difference in optimal objective value between baseline methods and our proposed C-MICL at $\alpha = 5\%$ across 100 optimization problems. Our approach shows a modest 8% difference compared to the method that achieved empirical coverage, demonstrating that C-MICL provides solution quality on par with other valid approaches. Lighter bars with an "x" indicate methods that did not meet the empirical feasibility target. Five outliers were removed for the W-MICL(50) Gradient Boosted Tree model, though the comparison remains statistically insignificant with or without them.

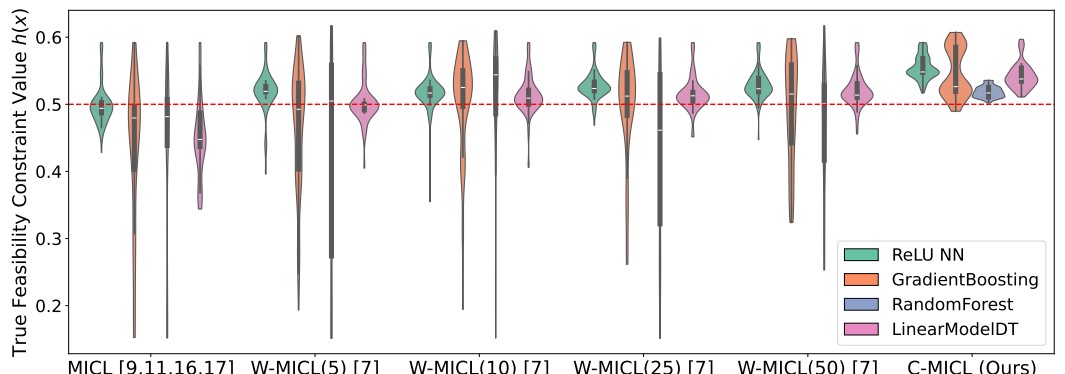

Figure 14: Violin plots of the true constraint values $h(x)$ for each method at $\alpha = 5\%$. The dotted line at $0.5$ denotes the lower bound enforced in the oracle constraint where values falling below this line indicate true infeasibilities. At this stricter confidence level, C-MICL maintains the highest reliability, producing the fewest and smallest violations across all methods.

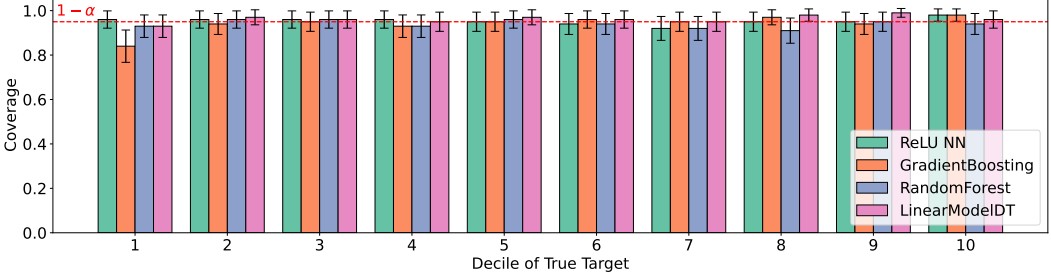

Figure 15: Empirical out-of-sample coverage at $\alpha = 5\%$ for all predictive models, evaluated on 1,000 test points and stratified by deciles of the true output $y$. Our conformal prediction sets achieve the desired coverage level across deciles. Notably, the ground-truth feasibility threshold ($y \geq 50$) lies in the 9th and 10th deciles, where all methods achieve valid empirical coverage, providing empirical support for the assumptions in Theorem 4.1.

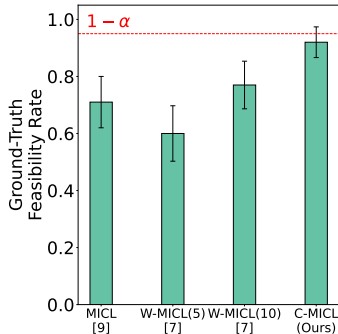 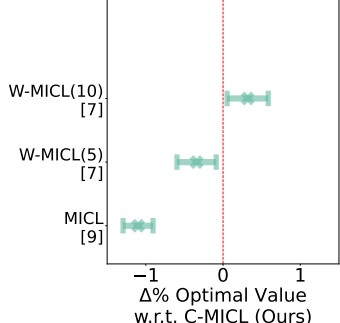 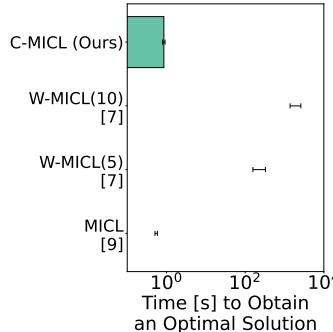

Figure 16: Empirical ground-truth feasibility rates across MICL methods on 100 optimization problem instances at $\alpha = 5\%$. C-MICL (rightmost bar) is the only method that consistently achieves the target feasibility threshold of at least 95%, in line with theoretical guarantees. All baseline methods fall short of this benchmark.

Figure 17: Relative difference in optimal objective value between baseline MICL methods and C-MICL across 100 optimization problem instances at $\alpha = 5\%$. The differences average about 1%, with all methods achieving statistically similar solution quality. However, none of the baselines produce implementable solutions, as indicated by the lighter bars, which represent methods that failed to meet the required empirical coverage guarantee. Only our method satisfies this guarantee.

Figure 18: Average time to compute an optimal solution for each MICL method on 100 optimization instances at $\alpha = 5\%$. C-MICL matches the runtime of single-model MICL and outperforms ensemble-based methods by a wide margin. Lighter bars represent approaches that did not achieve the target feasibility level.

## E.2 Classification

The following three figures present the performance of different MICL methods applied to the food basket design problem under a classification setting, with $\alpha = 5\%$. These figures assess the feasibility, objective value, and computational efficiency of the methods across 100 problem instances, highlighting the strengths of C-MICL in comparison to baseline approaches. Specifically, Figure 16 illustrates the empirical ground-truth feasibility rate, Figure 17 shows the relative difference in optimal objective value, and Figure 18 reports the average computational time required to obtain an optimal solution. Finally, Figures 19 and 20 present the empirical coverage achieved by each model across 1,000 out-of-sample points, grouped by the label $y$ category, for $\alpha = 10\%$ and $\alpha = 5\%$, respectively. These figures report the proportion of instances in which the true value of $y$ falls within the predicted conformal set. Our conformal sets empirically satisfy the ground-truth feasibility coverage guarantee from Lemma 3.1, achieving coverage rates of 88.22% (feasible) and 96.19% (infeasible) for $\alpha = 10\%$, and 93.41% (feasible) and 98.79% (infeasible) for $\alpha = 5\%$.

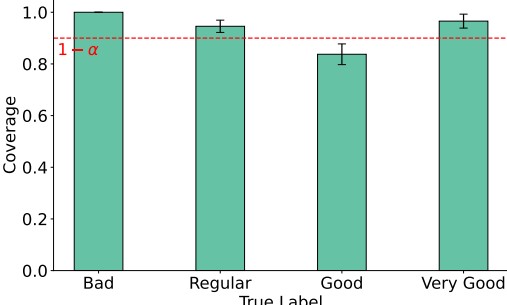
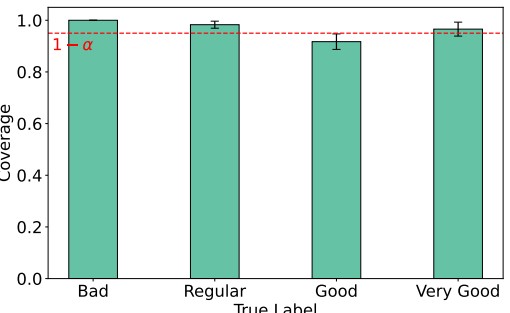

Figure 19: Empirical out-of-sample coverage, evaluated over 1,000 test points and stratified by the true categorical label of $y$ at $\alpha = 10\%$. Across all categories, the predicted conformal sets capture the true label at the desired coverage level. Notably, for the "Good" and "Very Good" categories, (which correspond to the ground-truth feasibility region) the models approximately achieve valid empirical coverage, lending support to the assumptions in Theorem 4.1.

Figure 20: Evaluation of empirical coverage with ReLU NNs over 1,000 test samples, grouped by the true $y$ label, for a target level of at $\alpha = 5\%$. The model produces conformal prediction sets that contain the true class label at the expected rate. In particular, the "Good" and "Very Good" categories, those relevant to enforcing feasibility, demonstrate approximate empirical coverage, providing evidence consistent with Theorem 4.1.

