# OpenReview forum: "Conformal Mixed-Integer Constraint Learning with Feasibility Guarantees"
_NeurIPS.cc/2025/Conference — NeurIPS 2025 spotlight_

### Official Review · Reviewer_o8nC · 2025-06-30

**Clarity:** 3
**Significance:** 3
**Originality:** 3
**Rating:** 4
**Confidence:** 3

**Summary:**

In this paper, the author proposes a new method to address challenges in constraint learning(CL) in optimization problems. The authors introduces a framework that integrates conformal prediction to CL problems to ensure that solutions to optimization problems with learned constraints are practically implementable.

**Questions:**

[Influence of the specific conformal scores] In Section 4.1 and Section 4.2, specific conformal scores are chosen for the regression and classification problems, respectively. A more detailed analysis of how the choice of a conformal score function influences the results would be a valuable addition. Furthermore, how to evaluate or select between different valid score functions for a given application?

**Ethical Concerns:**

["NO or VERY MINOR ethics concerns only"]

**Final Justification:**

The paper is interesting and the authors solve my concerns. I've changed my score accordingly.

**Quality:**

3

**Strengths And Weaknesses:**

Strengths:
-  The paper is well-written and well-organized.
-  The authors clearly define the problem, and the procedure to implement the proposed framework.
-  The authors validate their approach on both synthetic dataset and real dataset with real-world applications.


Weakness:
-  [Conditional independence assumption] The authors discuss the potential limitations of the assumption. While acknowledged as a necessary "theoretical compromise", the assumption is a notable one, and its potential impact warrants careful consideration by practitioners.

---

> ### Author Rebuttal · Authors · 2025-07-31
>
> Thank you for the thoughtful review and helpful feedback. Below we address weaknesses and questions.
>
> - W1: We thank the reviewer for highlighting the importance of Assumption 4.1 and for prompting a more detailed discussion of its role, interpretation, and limitations. We agree that this assumption is central to our theoretical guarantees and warrants careful justification. We believe it is reasonable in our setting and necessary to extend conformal guarantees to constrained optimization problems.
>
> Assumption 4.1 states that, conditional on ground-truth feasibility ($h(x) \in \mathcal{Y}$), the event of C-MICL feasibility ($(x, z) \in \mathcal{F}_N = \\{(x, z) \in \mathcal{X}: g(x, z) \leq 0,\ \mathcal{C}(x) \subseteq \mathcal{Y}\\}$) is independent of whether the conformal set contains the true function value ($h(x) \in \mathcal{C}(x)$).
>
> To motivate Assumption 4.1 and clarify when it is plausible, note first that the ground-truth feasibility (GTF) conditional coverage guarantee from Lemma 3.1 can be achieved in a fully data-driven way (e.g., using Mondrian conformal prediction or other label-conditional conformal methods):
> $$\mathbb{P}(h(x) \in \mathcal{C}(x) \mid h(x) \in \mathcal{Y}) \geq 1-\alpha$$
> $$\mathbb{P}(h(x) \in \mathcal{C}(x) \mid h(x) \notin \mathcal{Y}) \geq 1-\alpha$$
>
> However, in C-MICL we aim to guarantee coverage over the feasible region of the optimization problem $\mathcal{F}_N = \\{(x, z) \in \mathcal{X}: g(x, z) \leq 0,\ \mathcal{C}(x) \subseteq \mathcal{Y}\\}$, i.e.,
> $$\mathbb{P}(h(x) \in \mathcal{C}(x) \mid (x, z) \in \mathcal{F}_N) \geq 1-\alpha$$
>
> If the predictive model $\widehat{h}(x)$ were perfect (i.e., $\widehat{h}(x) = h(x)$), then the regions $\mathcal{F}_N$ and the ground-truth feasible region $\mathcal{F}$ would coincide, and the coverage guarantee would transfer directly. However, since we are interested in the more realistic case where $\widehat{h}(x)$ is imperfect, $\mathcal{F}_N$ and $\mathcal{F}$ differ in a data-dependent way.
>
> In this case, since the feasible region $\mathcal{F}_N$ is implicitly shaped by the calibration set (via $\mathcal{C}(x)$), there is a natural dependency between the feasible solutions of the C-MICL problem and the calibration data, which invalidates standard conformal guarantees relying on i.i.d. calibration and test data.
> Assumption 4.1 precisely seeks to decouple this dependency: it allows us to approximate conformal coverage within $\mathcal{F}_N$ by assuming that feasibility does not systematically bias conformal validity, once conditioned on ground-truth feasibility.
>
> To build intuition, consider partitioning $\mathcal{F}_N$ into two disjoint subsets: $\mathcal{F}_N \cap \mathcal{F}$ and $\mathcal{F}_N \cap \mathcal{F}^c$. Then, Assumption 4.1 implies that $\mathcal{F}_N \cap \mathcal{F}$ (respectively $\mathcal{F}_N \cap \mathcal{F}^c$) is not systematically biased towards a region of $\mathcal{F}$ ($\mathcal{F}^c$) that is miscalibrated. Mathematically,
> $$\mathbb{P}(h(x) \in \mathcal{C}(x) \mid (x, z) \in \mathcal{F}_N \cap \mathcal{F}) = \mathbb{P}(h(x) \in \mathcal{C}(x) \mid (x, z) \in \mathcal{F}_N, h(x) \in \mathcal{Y}) \approx \mathbb{P}(h(x) \in \mathcal{C}(x) \mid h(x) \in \mathcal{Y}) $$
> $$\mathbb{P}(h(x) \in \mathcal{C}(x) \mid (x, z) \in \mathcal{F}_N \cap \mathcal{F}^c) = \mathbb{P}(h(x) \in \mathcal{C}(x) \mid (x, z) \in \mathcal{F}_N, h(x) \notin \mathcal{Y}) \approx \mathbb{P}(h(x) \in \mathcal{C}(x) \mid h(x) \notin \mathcal{Y}) $$
> These enable us to translate the conformal coverage guarantees from the ground-truth feasible region $\mathcal{F}$ to the feasible set $\mathcal{F}_N$ used in the optimization.
>
> Assumption 4.1 is therefore reasonable when the calibration data adequately covers the parts of the input space that intersect the feasible region $\mathcal{F}_N$, both within the ground-truth feasible region $\mathcal{F}$ and its complement $\mathcal{F}^c$. In this sense, it aligns with standard generalization assumptions that require the training and calibration data to be representative of the regions where predictions are deployed. Alternatively, Assumption 4.1 can be approximated using more granular conditional conformal methods, by partitioning the optimization region into finer subregions and enforcing local coverage guarantees within each, which can then be translated to the feasible set $\mathcal{F}_N$.
>
> We will revise the paper to better explain this interpretation and expand the discussion on when the assumption might fail. For instance, the assumption may break down if the feasible region $\mathcal{F}_N$ is heavily concentrated in areas where the calibration set is sparse or systematically miscalibrated. In our experimental settings, we observe good empirical alignment between target and achieved coverage (Appendix E), suggesting that Assumption 4.1 holds reasonably well in practice in realistic data scenarios.
>
>
> - Q1: Thanks for the opportunity to provide more details on the influence of the specific conformal scores used in our C-MICL framework. In the current work, conformal score functions were selected to maintain general applicability across a wide range of regression and classification models, in line with the model-agnostic nature of the proposed C-MICL framework. Following standard practice in the conformal prediction literature, we employed score functions that are symmetric and proportional to the model's estimated uncertainty (see Angelopoulos et al. [52]). By aligning with widely accepted strategies in the literature, we aimed to provide a robust yet general approach that performs well across various learning settings without relying on model-specific scores.
>
> Specifically, in our experiments  we use a residual-based score function for the regression setting, as it relies solely on point predictions and can therefore be applied uniformly across all regression models. For classification, the score function was chosen to preserve linearity in the decision space, enabling efficient integration into the underlying optimization problem.
>
> We agree that exploring the effect of different valid conformal scores and providing guidance for their selection is a valuable direction for future work. There is already a substantial body of literature that investigates alternative score functions and their impact on the resulting conformal sets, which we will add to the revised manuscript. These works highlight the importance of constructing informative conformal sets to ensure both the feasibility and the quality of the downstream optimization problems. We appreciate the opportunity to discuss this further and will incorporate this perspective in the updated manuscript.

---

> > ### Comment · Reviewer_o8nC · 2025-08-04
> >
> > Thank you for addressing my concern. I will increase my rating accordingly.

---

### Official Review · Reviewer_zD5K · 2025-07-01

**Clarity:** 3
**Significance:** 2
**Originality:** 3
**Rating:** 4
**Confidence:** 3

**Summary:**

This paper introduces Conformal Mixed-Integer Constraint Learning (C-MICL), a novel framework that integrates conformal prediction into mixed-integer constraint learning (MICL) to ensure probabilistic feasibility guarantees. The core idea is to replace heuristic approaches (e.g., ensemble-based W-MICL) with a principled conformal prediction-based mechanism that enables certified feasibility of solutions to learned-constraint optimization problems, under a mild conditional independence assumption. The method is model-agnostic and MIP-compatible, requiring only that the learned constraint model is representable as a mixed-integer program. It supports both regression and classification settings. Theoretical results are provided to justify the probabilistic guarantees. The empirical evaluation spans real-world-inspired MILP problems, including a chemical reactor optimization and a food basket design task. C-MICL consistently achieves the desired feasibility level, offers competitive objective performance, and exhibits significant speedups compared to ensemble methods.

**Questions:**

1.	As stated in the Limitation, in cases where the base model is inaccurate or uncertainty is large, the conformal set may become too wide, potentially making the C-MICL problem infeasible. Do the authors have strategies for detecting or mitigating this issue (e.g., fallback mechanisms, feasibility repair, or relaxed constraints)?
2.	While the paper claims generality beyond MILPs, experiments focus exclusively on linear settings. Have the authors tested C-MICL in mixed-integer nonlinear problems (MINLPs)? If so, what challenges arise?

**Ethical Concerns:**

["NO or VERY MINOR ethics concerns only"]

**Final Justification:**

The paper proposes a novel conformal mixed-integer constraint learning. During the author-reviewer discussion, the authors provided detailed response, which addressed my concerns. So I will maintain my score that is inclined to acceptance.

**Limitations:**

The authors have provided a thoughtful discussion of the limitations in Section 6, notably including the reliance on the conditional independence assumption, and the potential for conformal sets to become overly conservative when the predictive model is poorly calibrated. These are critical aspects and are appropriately acknowledged.

**Quality:**

3

**Strengths And Weaknesses:**

Strengths:
1. The proposed C-MICL framework provides formal probabilistic guarantees on the feasibility of solutions with respect to the unknown true constraints, which is a major advance over existing heuristic approaches.

2. C-MICL avoids ensemble-based methods and scales efficiently—requiring only one or two models—while achieving significant computational speedups without sacrificing objective performance.

3. Across diverse case studies, C-MICL consistently achieves target feasibility levels, while maintaining competitive objective values.

4. The paper is well-organized and clearly written, with intuitive explanations, theoretical results, and implementation details presented in a logical and accessible manner.

Weaknesses:
1. While plausible, the assumption that C-MICL feasibility and conformal coverage are conditionally independent (Assumption 4.1) is strong and difficult to validate in practice. This is the linchpin of the main guarantee, and its empirical verification is only approximate.

2. In cases of poorly performing base predictors or high noise, conformal sets may become overly conservative, potentially leading to empty or trivial feasible sets. While discussed in the limitations, this could be a practical barrier.

---

> ### Author Rebuttal · Authors · 2025-07-31
>
> Thank you for the thoughtful review and helpful feedback. Below we address weaknesses and questions.
>
> - W1: We thank the reviewer for highlighting the importance of Assumption 4.1 and for prompting a more detailed discussion of its role, interpretation, and limitations. We agree that this assumption is central to our theoretical guarantees and warrants careful justification. We believe it is reasonable in our setting and necessary to extend conformal guarantees to constrained optimization problems.
>
> Assumption 4.1 states that, conditional on ground-truth feasibility ($h(x) \in \mathcal{Y}$), the event of C-MICL feasibility ($(x, z) \in \mathcal{F}_N = \\{(x, z) \in \mathcal{X}: g(x, z) \leq 0,\ \mathcal{C}(x) \subseteq \mathcal{Y}\\}$) is independent of whether the conformal set contains the true function value ($h(x) \in \mathcal{C}(x)$).
>
> To motivate Assumption 4.1 and clarify when it is plausible, note first that the ground-truth feasibility (GTF) conditional coverage guarantee from Lemma 3.1 can be achieved in a fully data-driven way (e.g., using Mondrian conformal prediction or other label-conditional conformal methods):
> $$\mathbb{P}(h(x) \in \mathcal{C}(x) \mid h(x) \in \mathcal{Y}) \geq 1-\alpha$$
> $$\mathbb{P}(h(x) \in \mathcal{C}(x) \mid h(x) \notin \mathcal{Y}) \geq 1-\alpha$$
>
> However, in C-MICL we aim to guarantee coverage over the feasible region of the optimization problem $\mathcal{F}_N = \\{(x, z) \in \mathcal{X}: g(x, z) \leq 0,\ \mathcal{C}(x) \subseteq \mathcal{Y}\\}$, i.e.,
> $$\mathbb{P}(h(x) \in \mathcal{C}(x) \mid (x, z) \in \mathcal{F}_N) \geq 1-\alpha$$
>
> If the predictive model $\widehat{h}(x)$ were perfect (i.e., $\widehat{h}(x) = h(x)$), then the regions $\mathcal{F}_N$ and the ground-truth feasible region $\mathcal{F}$ would coincide, and the coverage guarantee would transfer directly. However, since we are interested in the more realistic case where $\widehat{h}(x)$ is imperfect, $\mathcal{F}_N$ and $\mathcal{F}$ differ in a data-dependent way.
>
> In this case, since the feasible region $\mathcal{F}_N$ is implicitly shaped by the calibration set (via $\mathcal{C}(x)$), there is a natural dependency between the feasible solutions of the C-MICL problem and the calibration data, which invalidates standard conformal guarantees relying on i.i.d. calibration and test data.
> Assumption 4.1 precisely seeks to decouple this dependency: it allows us to approximate conformal coverage within $\mathcal{F}_N$ by assuming that feasibility does not systematically bias conformal validity, once conditioned on ground-truth feasibility.
>
> To build intuition, consider partitioning $\mathcal{F}_N$ into two disjoint subsets: $\mathcal{F}_N \cap \mathcal{F}$ and $\mathcal{F}_N \cap \mathcal{F}^c$. Then, Assumption 4.1 implies that $\mathcal{F}_N \cap \mathcal{F}$ (respectively $\mathcal{F}_N \cap \mathcal{F}^c$) is not systematically biased towards a region of $\mathcal{F}$ ($\mathcal{F}^c$) that is miscalibrated. Mathematically,
> $$\mathbb{P}(h(x) \in \mathcal{C}(x) \mid (x, z) \in \mathcal{F}_N \cap \mathcal{F}) = \mathbb{P}(h(x) \in \mathcal{C}(x) \mid (x, z) \in \mathcal{F}_N, h(x) \in \mathcal{Y}) \approx \mathbb{P}(h(x) \in \mathcal{C}(x) \mid h(x) \in \mathcal{Y}) $$
> $$\mathbb{P}(h(x) \in \mathcal{C}(x) \mid (x, z) \in \mathcal{F}_N \cap \mathcal{F}^c) = \mathbb{P}(h(x) \in \mathcal{C}(x) \mid (x, z) \in \mathcal{F}_N, h(x) \notin \mathcal{Y}) \approx \mathbb{P}(h(x) \in \mathcal{C}(x) \mid h(x) \notin \mathcal{Y}) $$
> These enable us to translate the conformal coverage guarantees from the ground-truth feasible region $\mathcal{F}$ to the feasible set $\mathcal{F}_N$ used in the optimization.
>
> Assumption 4.1 is therefore reasonable when the calibration data adequately covers the parts of the input space that intersect the feasible region $\mathcal{F}_N$, both within the ground-truth feasible region $\mathcal{F}$ and its complement $\mathcal{F}^c$. In this sense, it aligns with standard generalization assumptions that require the training and calibration data to be representative of the regions where predictions are deployed. Alternatively, Assumption 4.1 can be approximated using more granular conditional conformal methods, by partitioning the optimization region into finer subregions and enforcing local coverage guarantees within each, which can then be translated to the feasible set $\mathcal{F}_N$.
>
> We will revise the paper to better explain this interpretation and expand the discussion on when the assumption might fail. For instance, the assumption may break down if the feasible region $\mathcal{F}_N$ is heavily concentrated in areas where the calibration set is sparse or systematically miscalibrated. In our experimental settings, we observe good empirical alignment between target and achieved coverage (Appendix E), suggesting that Assumption 4.1 holds reasonably well in practice in realistic data scenarios.
>
>
> - W2 + Q1: We thank the reviewer for raising this important question and for the opportunity to elaborate on this limitation of the C-MICL framework. As noted in the manuscript,  the optimization problem may become infeasible when the conformal sets are overly conservative (e.g., too wide or trivial), which can  occur due to a poorly performing base predictor or high intrinsic noise. In practice, this issue can be diagnosed by evaluating the predictive performance of $\widehat{h}(x)$, analyzing the distribution of conformal scores on calibration data, and inspecting the  conformal sets on a hold-out test set.
>
> A natural way to mitigate this issue is to improve the underlying predictive model using standard machine learning techniques or to refine the choice of conformal score functions to better capture uncertainty. However, a less obvious but effective approach is to reduce the statistical confidence level used to construct $\mathcal{C}(x)$. This leads to smaller conformal sets and can restore feasibility, while still providing explicit and interpretable probabilistic guarantees at a potentially lower coverage level.
>
> In our experiments, we find that with reasonably well-trained models the conformal sets remain informative and the C-MICL problem is typically feasible in practice. Finally, we also note that this limitation is not unique to C-MICL. For instance, MICL and W-MICL may be more sensitive to poor model quality, as they rely solely on point predictions without accounting for uncertainty, which can result in solutions that are not practically implementable. We will clarify this point further in the revised version and thank the reviewer again for highlighting this issue.
>
>
> - Q2: We appreciate the reviewer’s attention to the scope of our C-MICL method. As correctly noted, our theoretical results do not assume linearity in the underlying optimization problems and extend naturally to mixed-integer nonlinear programs. We focused on MILPs in our experiments to ensure global optimality across all methods (MICL, W-MICL, C-MICL) tractably, allowing for fair comparisons and avoiding confounding due to local optima. Our extensive evaluation setup involved solving each method-model pair across hundreds of independent instances, resulting in several hundred globally solved problems per approach. This level of evaluation is considerably more tractable in the MILP setting.
>
> That said, our C-MICL approach applies directly to MINLPs, and our theory continues to provide probabilistic ground-truth feasibility guarantees. The primary challenge in extending the experimental evaluation to MINLPs is the significant increase in computational effort required to solve these problems to global optimality. We will include this discussion in the updated version of the article, along with a discussion of the specific computational challenges that arise when applying C-MICL to MINLPs.

---

> > ### Comment · Reviewer_zD5K · 2025-08-04
> > **Reply**
> >
> > Thanks for the authors' rebuttal. I will maintain my original score.

---

### Official Review · Reviewer_fgyD · 2025-07-01

**Clarity:** 3
**Significance:** 3
**Originality:** 3
**Rating:** 4
**Confidence:** 3

**Summary:**

The authors discuss mathematical optimization problems for which a part of the constraint functions are not explicitly known. These unknown functions are replaced with predictive models using data. Although this idea has been studied in the literature before, the authors address the concern about the feasibility of the obtained optimal solutions, since these solutions may only satisfy the predictive model but not necessarily the true (unknown) mapping.

Using the core results from conformal prediction theory, the authors propose, what they call, Conformal Mixed-Integer Constraint Learning (C-MICL) method, which gives probabilistic guarantees on the learned constraint function and works with any model that can be represented as a mixed-integer program. To this end, the authors integrate conformal predictions in both a regression and a classification setting to ensure ground-truth feasibility with a certain probability. The authors compare their method on different metrics against multiple benchmarks on regression and classification datasets.

**Questions:**

- What happens when the feasible set for the outcome, Y is more complicated; e.g., it is defined by using integer variables and constraints?
- Related to previous question, would the proposed approach still work if g function in (MICL) model also involves y?
- In several places, the authors call the proposed approach model-agnostic. However, it does require predictive models to be MIP-representable, doesn't it?
- In the experiments section, there are error bars, but it is not mentioned in the section what these bars represent exactly. Could the author elaborate on this when discussing the plots in the text? Moreover,  for the C-MICL method, the reported ground-truth feasibility rate does not always meet the target coverage level, as the error bars suggest. Could you elaborate why this happens, because the theory would suggest the target coverage level should be met? Is it because you are talking about empirical errors? Can the authors elaborate?
- The authors claim Scalability and Efficiency concerning the size of the dataset. However, in your experiments, there seems to be a comparison with benchmarks for a fixed size of the dataset. For this dataset, the C-MICL method seems more efficient, and as it needs fewer models to train, the claim of scalability seems valid. However, there is no experimental support that this will hold for different dataset sizes, as with a fixed split for training and conformal calibration, the size of the dataset would affect all methods. Is this observation correct? Did the authors make comparisons for different dataset sizes? If so, could the authors elaborate on their claim regarding scalability?
- What is the effect of the size of the data set used for conformal calibration? In one experiment, the authors use 20% of the data, and in the other, the authors use 8%. What is common practice here? How did you derive these different split ratios? And what effect would the size of the data set used for conformal calibration have on the discussed measures?

**Ethical Concerns:**

["NO or VERY MINOR ethics concerns only"]

**Final Justification:**

After reading through other referees' comments and the authors' responses, I have decided to maintain my score.

**Limitations:**

Yes.

**Paper Formatting Concerns:**

No concerns.

**Quality:**

3

**Strengths And Weaknesses:**

Strengths:
- The paper is well-written. The overall message is clear, the mathematics is sound and well-defined.
- Results look promising: Although other methods tend to find better optimal values, there is a clear improvement in ground-truth feasibility rate, and runtime when using the C-MICL method.

Weaknesses:
- Discussion on the conditional independence assumption is brief. The authors mention in the discussion that this assumption is reasonable. I would like to know earlier why this is a reasonable assumption and what situations this assumption does not hold.
- All of a sudden, a secondary regression model comes into the picture in Section 4.1. It is not clear to understand its role, and why is it needed.
- The authors report only the rate of the ground truth feasibility (I guess this is the number of feasible solutions among 100 instances). However, there is no evaluation of the magnitude of infeasibility. There could be many solutions that are obtained with other approaches which are very close to be feasible region, whereas C-MICL may result in a few but grossly infeasible solutions.

---

> ### Author Rebuttal · Authors · 2025-07-31
>
> Thank you for the thoughtful review and helpful feedback. Below we address weaknesses and questions.
>
> - W1: We thank the reviewer for highlighting the importance of Assumption 4.1. We agree that this assumption is central to our theoretical guarantees and warrants careful justification.
>
> Intuitively, Assumption 4.1 is plaussible when the calibration data adequately covers the parts of the input space that intersect the feasible region $\mathcal{F}_N$, both within the ground-truth feasible region $\mathcal{F}$ and its complement $\mathcal{F}^c$. In this sense, it aligns with standard generalization assumptions that require the training and calibration data to be representative of the regions where predictions are deployed. Alternatively, Assumption 4.1 can be approximated in a data-driven way using more granular conditional conformal methods, by partitioning the optimization region into finer subregions and enforcing local coverage guarantees within each, which can then be translated to the feasible set $\mathcal{F}_N$.
>
> We will revise the paper to better explain this interpretation and expand the discussion on when the assumption might fail. For instance, the assumption may break down if the feasible region $\mathcal{F}_N$ is heavily concentrated in areas where the calibration set is sparse or systematically miscalibrated. In our experimental settings, we observe good empirical alignment between target and achieved coverage (Appendix E), suggesting that Assumption 4.1 holds reasonably well in practice in realistic data scenarios.
>
> - W2: The secondary regression model $\hat{u}(x)$ introduced in Section 4.1 is used to quantify the uncertainty around the predictive model $\hat{h}(x)$. This secondary model is essential for constructing adaptive conformal sets whose size depends on the input $x$. As is standard in the conformal prediction literature, such adaptive sets allow us to maintain valid coverage while tailoring the prediction interval to the local uncertainty of the model (see Angelopoulos et al [52] and Papadopoulos et al. [56]). The use of a separate model for estimating uncertainty is a common and well-established practice in conformal regression, and we will revise the manuscript to better introduce and motivate the role of $\hat{u}(x)$ when it is first presented.
>
> - W3: Thank you for raising this valid and insightful concern about violation magnitude. We evaluated the magnitude of feasibility violantions to check whether C-MICL produces grossly infeasible solutions. Specifically, we inspected the actual values of the constraint violations and confirmed that, even when violations occurred, they were relatively minor. For example, in the NN classification case with $\alpha = 10\%$, which was the scenario with the highest number of violations for C-MICL (11 out of 100), the true values were: [0.395, 0.424, 0.455, 0.462, 0.479, 0.482, 0.487, 0.489, 0.489, 0.493, 0.495], under a feasibility threshold of 0.5. These values show that C-MICL tends to stay close to the feasible region even when feasibility is not strictly satisfied. We appreciate the suggestion and will incorporate violin plots to the Appendix in the revised manuscript to visualize the distribution of constraint violations across methods and scenarios.
>
> - Q1: The C-MICL approach remains applicable when the feasible set $\mathcal{Y}$ is more complicated, for instance, involving integer variables and constraints. While feasibility checking may appear more challenging when $\mathcal{Y}$ is more complex, the modeling effort lies in formulating the feasible set $\mathcal{Y}$ as a MIP, which is inherent to any constraint-learning method that must verify feasibility with respect to $\mathcal{Y}$, not specific to our proposed conformal approach. These MIP-representable constraint sets include finite unions of intervals, discrete sets, and sets defined by (non)linear inequalities, piecewise (non)linear constraints, and logical operations (e.g, disjunctions), capturing a wide class of practical applications in operations research and optimization (see "50 Years of Mixed-Integer Nonlinear and Disjunctive Programming" by Kronqvist et al.).
>
> In such cases, C-MICL still enforces feasibility by verifying that the conformal prediction set lies entirely within the feasible set, i.e., $\mathcal{C}(x) \subseteq \mathcal{Y}$. This condition remains consistent with our formulations  in Equations (4) and (6) for regression and classification, respectively.  We appreciate the opportunity to discuss this point and will clarify the generality of our method subject to $\mathcal{Y}$ being MIP-representable in the revised manuscript.
>
> - Q2: Thank you for this interesting question. When the $g$ function involves the outcome variable $y$, the conformalization procedure becomes more complex. Since we have a conformal prediction set $\mathcal{C}(x)$ for $y = \widehat{h}(x)$, evaluating $g$ requires considering all possible values in this set, yielding a corresponding prediction set $\mathcal{G}(x,z) = \\{g(x, \widehat{y}, z) : \widehat{y} \in \mathcal{C}(x)\\}$. The constraint $g(x,z) \leq 0$ in the MICL formulation becomes $\sup \mathcal{G}(x,z) \leq 0$, which remains MIP-representable and can be incorporated in our framework.
>
> However, extending the coverage guarantees from $\mathcal{C}(x)$ to the prediction set $\mathcal{G}(x,z)$ is non-trivial for arbitrary functions $g$. The challenge lies in preserving the coverage properties when the conformal sets are transformed through potentially complex nonlinear functions. While this extension is theoretically possible under certain regularity conditions on $g$, it requires careful analysis of how prediction uncertainties propagate through the constraint function. We agree this is a promising direction for future research and will add this discussion to the updated manuscript.
>
> - Q3: Thank you for this valuable observation. You are correct that our framework requires the predictive model to be MIP‑representable. When we describe the approach as "model‑agnostic", we mean that it can accommodate any predictive model for which a MIP formulation exists. In particular, many widely used machine learning models can be formulated as mixed-integer programs  in practice. Examples include linear, polynomial, and symbolic regression models (Wilson et al.), support vector machines (Maragno et al. [7]), and tree ensembles (Mistry et al. [11]). Moreover, several neural network architectures also admit MIP representations, including feedforward networks (Fischetti et al. [8]), graph neural networks (see Hojny et al.), and Kolmogorov–Arnold Networks (Karia et al.). Thus, although we impose the MIP-representability requirement, the framework remains broadly applicable in real‑world settings. We note that this distinction is noted in Remark 3.1 on page 4. Nonetheless, we are happy to clarify the intended meaning of “model‑agnostic” in the revised manuscript.
>
> - Q4: The error bars correspond to 95\% confidence intervals for: (i) estimated ground-truth feasibility rates in Figures 1 and 4, (ii) average relative differences in objective values in Figures 2 and 5, and (iii) average computational times in Figures 3 and 6. All estimates are computed over 100 randomly generated optimization instances. Specific formulas for these confidence intervals are provided in Appendix E, and we will add this clarification to the figure captions and when discussing the plots in the text.
>
> Regarding the reported ground-truth feasibility of the C-MICL method, the observed variations around the target coverage level stem from finite sample effects rather than theoretical violations. Our estimates use 100 independent optimization instances, introducing natural estimation uncertainty. Crucially, C-MICL contains the nominal level within the confidence intervals across all experiments, indicating no statistically significant deviation (miscoverage) from theoretical guarantees.
>
> - Q5: As correctly noted, our claim of efficiency primarily refers to the fact that C-MICL requires training at most two predictive models, in contrasts to conformal ensemble-based methods which often involve training many models, leading to optimization times that are orders of magnitude longer.
>
> With respect to scalability in dataset size, we clarify that our claim refers specifically to the  optimization phase. Existing methods, such as that of Zhao et al. [47], embed the entire training dataset as explicit constraints in the optimization model. This results in a formulation whose size grows linearly with the number of training samples, substantially affecting scalability and tractability of the optimization problem. In contrast, our framework trains predictive models offline, and once trained, the optimization problem is independent of the training dataset size. This means that increasing the size of the training data does not impact the tractability or runtime of the optimization phase. For this reason, we did not test different dataset sizes to assess optimization time, as it remains fixed once the models are trained. We appreciate the reviewer’s comment and will revise the manuscript to clarify this distinction more explicitly.
>
> - Q6: Thank you for the thoughtful question. We chose the calibration split ratios to ensure approximately 200 data points in the conformal calibration set, following common practice in the literature, which typically recommends using between 100 and 500 points (see Angelopolous et al. [52]). Naturally, there is a trade-off involved: allocating more data to the calibration set improves the accuracy of the conformal quantile estimation, while reducing the amount of training data may affect the quality of the prediction model. Our choice aims to reflect realistic data limitations while demonstrating that the C-MICL approach remains effective in such settings. We will clarify this rationale and discuss the trade-off more explicitly in the revised manuscript.

---

> > ### Comment · Reviewer_fgyD · 2025-08-05
> > **Clear response and well-made case**
> >
> > The response of the authors is clear, and they made their case well. I thank them for their efforts. After careful consideration and taking into account the views of other reviewers, I am leaning toward keeping the score as it is.

---

### Official Review · Reviewer_Pih6 · 2025-07-03

**Clarity:** 4
**Significance:** 3
**Originality:** 3
**Rating:** 5
**Confidence:** 4

**Summary:**

This paper proposes a methodology, called Conformal Mixed-Integer Constraint Learning (C-MICL), whereby conformal prediction is used to obtain probabilistic guarantees of feasibility on the solution to optimization problems with unknown constraints (where data can be used to learn the constraint). In particular, when the decision variables $x$ of an optimization problem must satisfy the constraint $h(x) \in \mathcal{Y}$ for unknown $h$, the authors propose to construct a conformal prediction set $\mathcal{C}(x)$ which, under standard exchangeability assumptions, contains the value $h(x)$ with probability $1-\alpha$ for some chosen $\alpha$. Then, the set containment constraint $\mathcal{C}(x) \subseteq \mathcal{Y}$ can be used as a (conservative, but valid with high probability) replacement for the original unknown constraint, and the the feasible set of the new problem will be feasible for the original constraint with probability $1-\alpha$. The authors describe how this framework can be used in both the regression and classification settings, and run extensive experiments demonstrating that their approach yields better feasibility rate when compared to previous methods, while remaining computationally tractable.

**Questions:**

- The conformal set containment constraint $\mathcal{C}(x) \subseteq \mathcal{Y}$ seems like a challenging constraint to enforce in general, depending on the structure of $\mathcal{C}(x)$. In Sections 4.1 and 4.2 of the paper, you show how various regression and classification settings can yield conformal sets whose containment problems are MIP-representable, but are there other settings where such a formulation will not be easily obtained, and where this constraint will be hard to enforce in a MIP?
- If I understand correctly, assumption 4.1 is saying you assume that, if $h(x) \in \mathcal{Y}$, then the event $\mathcal{C}(x) \subseteq \mathcal{Y}$ is independent of the event $h(x) \in \mathcal{C}(x)$. Whether or not this assumption holds seems critical for the validity of the probabilistic feasibility guarantees. Is this a reasonable assumption in practice? If so, can you give formal, mathematical intuition for why this is an assumption that is likely to hold? I am willing to raise my score if provided a convincing answer to this question.
- The statement in lines 199-200 that "we assume that whether a point satisfies the constraint $\mathcal{C}(x) \subseteq \mathcal{Y}$ does not affect the probability that the conformal set contains the true function value" seems imprecise, given that the exact place where this is invoked in the proof includes the entire feasible set $\mathcal{F}_N$ of the C-MICL problem, not just the set containment $\mathcal{C}(x) \subseteq \mathcal{Y}$.
- It seems that replacing the constraint $h(x) \in \mathcal{Y}$ with the conformal set containment $\mathcal{C}(x) \subseteq \mathcal{Y}$ is conservative, as it results in larger objective values (seen in Figures 2 and 5). How do these values compare with the ground truth problem's optimal objective value?

**Ethical Concerns:**

["NO or VERY MINOR ethics concerns only"]

**Final Justification:**

The authors have sufficiently addressed my concerns about Assumption 4.1.

**Limitations:**

Yes

**Paper Formatting Concerns:**

Line 110 - "does" should be "do" to agree with the plural tense of "$z$ are" earlier in the sentence

**Quality:**

3

**Strengths And Weaknesses:**

Strengths:
- This is a very interesting problem setting, and the application of conformal prediction to this problem is a valuable insight.
- The paper is clearly written and easy to follow.
- The experiments are extensive and well-documented.

Weaknesses:
- There are a number of papers in the recent literature that have looked at integrating conformal prediction into various forms of optimization problems to deal with robust constraints and/or objectives. In the related work, the authors cite references [46] and [47], but it might be worth discussing the current work in the context of the broader set of papers at this intersection, including, e.g.:
  1. Johnstone and Cox, [Conformal Uncertainty Sets for Robust Optimization](https://arxiv.org/abs/2105.14957)
  2. Kiyani et al., [Decision Theoretic Foundations for Conformal Prediction: Optimal Uncertainty Quantification for Risk-Averse Agents](https://arxiv.org/abs/2502.02561)
  3. Patel et al., [Conformal Contextual Robust Optimization](https://arxiv.org/abs/2310.10003)
  4. Yeh et al., [End-to-End Conformal Calibration for Optimization Under Uncertainty](https://arxiv.org/abs/2409.20534)

- Assumption 4.1, on the conditional independence of the feasibility of C-MICL and conformal coverage, seems strong, and not obviously true.
- In cases where the function $h$ itself is not random, replacing the constraint $h(x) \in \mathcal{Y}$ with the conformal set containment $\mathcal{C}(x) \subseteq \mathcal{Y}$ seems like it would be conservative in general.

---

> ### Author Rebuttal · Authors · 2025-07-31
>
> Thank you for the thoughtful review and helpful feedback. Below we address weaknesses and questions.
>
> - W1: We thank the reviewer for highlighting these valuable contributions at the intersection of conformal prediction and optimization. We agree they provide important context and will include them in the revised manuscript. While Johnstone & Cox (2021) construct conformal ellipsoids for robust optimization, and Yeh et al. (2023) learn convex uncertainty sets for robust objectives using differentiable conformal layers, C-MICL focuses on conformalizing feasibility regions rather than robustifying objective values. Kiyani et al. (2023) and Patel et al. (2022) use conformal sets for utility-aware or risk-sensitive decision-making, optimizing value-at-risk or ensuring robust output quality. In contrast, C-MICL targets constraint satisfaction under model misspecification, supporting general nonconvex feasible regions and offering distribution-free feasibility guarantees. This leads to a distinct formulation centered on feasibility-aware decision-making rather than robustifying objectives/actions for downstream optimization problems. We thank the reviewer again for pointing out this additional work at the intersection of conformal prediction and optimization. We will include and discuss these additional references in the related work section in the revised manuscript.
>
> - Q1: We thank the reviewer for this important question. The key insight is that the conformal prediction sets $\mathcal{C}(x)$ used in C-MICL have predictable, well-structured forms that are always MIP-representable, making the containment constraint $\mathcal{C}(x) \subseteq \mathcal{Y}$ tractable regardless of the underlying prediction model.
>
> In the regression setting, the conformal set $\mathcal{C}(x)$ is always an interval of the form $\left[\hat{h}(x) \pm \hat{q}_{1-\alpha}(x)\right]$. Therefore, the containment constraint $\mathcal{C}(x) \subseteq \mathcal{Y}$ simplifies to checking whether this interval lies entirely within the feasible set $\mathcal{Y}$. When $\mathcal{Y} = [\underline{y}, \bar{y}]$, this reduces to two linear inequalities, which are directly representable in a MIP.
>
> In the classification setting, the conformal set $\mathcal{C}(x)$ is always a finite subset of the label space $\mathcal{K}$, and can be captured using binary indicator variables $w_k \in {0,1}$ for each class $k \in \mathcal{K}$. The constraint $\mathcal{C}(x) \subseteq \mathcal{Y}$ then becomes a set of linear conditions over these binary variables.
>
> Since conformal sets in C-MICL preserve these structured forms across different prediction models and conformity scores (intervals for regression, subsets for classification), the containment constraint remains computationally tractable, with detailed MIP formulations for both settings in Appendix C. We appreciate the opportunity to clarify this point and will emphasize in the revised manuscript the structured nature of conformal sets.
>
> - W2 + Q2: We thank the reviewer for highlighting the importance of Assumption 4.1 and for prompting a more detailed discussion of its role, interpretation, and limitations. We agree that this assumption is central to our theoretical guarantees and warrants careful justification. We believe it is reasonable in our setting and necessary to extend conformal guarantees to constrained optimization problems.
>
> Assumption 4.1 states that, conditional on ground-truth feasibility ($h(x) \in \mathcal{Y}$), the event of C-MICL feasibility ($(x, z) \in \mathcal{F}_N = \\{(x, z) \in \mathcal{X}: g(x, z) \leq 0,\ \mathcal{C}(x) \subseteq \mathcal{Y}\\}$) is independent of whether the conformal set contains the true function value ($h(x) \in \mathcal{C}(x)$).
>
> To motivate Assumption 4.1 and clarify when it is plausible, note first that the ground-truth feasibility (GTF) conditional coverage guarantee from Lemma 3.1 can be achieved in a fully data-driven way (e.g., using Mondrian conformal prediction or other label-conditional conformal methods):
> $$\mathbb{P}(h(x) \in \mathcal{C}(x) \mid h(x) \in \mathcal{Y}) \geq 1-\alpha$$
> $$\mathbb{P}(h(x) \in \mathcal{C}(x) \mid h(x) \notin \mathcal{Y}) \geq 1-\alpha$$
>
> However, in C-MICL we aim to guarantee coverage over the feasible region of the optimization problem $\mathcal{F}_N = \\{(x, z) \in \mathcal{X}: g(x, z) \leq 0,\ \mathcal{C}(x) \subseteq \mathcal{Y}\\}$, i.e.,
> $$\mathbb{P}(h(x) \in \mathcal{C}(x) \mid (x, z) \in \mathcal{F}_N) \geq 1-\alpha$$
>
> If the predictive model $\widehat{h}(x)$ were perfect (i.e., $\widehat{h}(x) = h(x)$), then the regions $\mathcal{F}_N$ and the ground-truth feasible region $\mathcal{F}$ would coincide, and the coverage guarantee would transfer directly. However, since we are interested in the more realistic case where $\widehat{h}(x)$ is imperfect, $\mathcal{F}_N$ and $\mathcal{F}$ differ in a data-dependent way.
>
> In this case, since the feasible region $\mathcal{F}_N$ is implicitly shaped by the calibration set (via $\mathcal{C}(x)$), there is a natural dependency between the feasible solutions of the C-MICL problem and the calibration data, which invalidates standard conformal guarantees relying on i.i.d. calibration and test data.
> Assumption 4.1 precisely seeks to decouple this dependency: it allows us to approximate conformal coverage within $\mathcal{F}_N$ by assuming that feasibility does not systematically bias conformal validity, once conditioned on ground-truth feasibility.
>
> To build intuition, consider partitioning $\mathcal{F}_N$ into two disjoint subsets: $\mathcal{F}_N \cap \mathcal{F}$ and $\mathcal{F}_N \cap \mathcal{F}^c$. Then, Assumption 4.1 implies that $\mathcal{F}_N \cap \mathcal{F}$ (respectively $\mathcal{F}_N \cap \mathcal{F}^c$) is not systematically biased towards a region of $\mathcal{F}$ ($\mathcal{F}^c$) that is miscalibrated. Mathematically,
> $$\mathbb{P}(h(x) \in \mathcal{C}(x) \mid (x, z) \in \mathcal{F}_N \cap \mathcal{F}) = \mathbb{P}(h(x) \in \mathcal{C}(x) \mid (x, z) \in \mathcal{F}_N, h(x) \in \mathcal{Y}) \approx \mathbb{P}(h(x) \in \mathcal{C}(x) \mid h(x) \in \mathcal{Y}) $$
> $$\mathbb{P}(h(x) \in \mathcal{C}(x) \mid (x, z) \in \mathcal{F}_N \cap \mathcal{F}^c) = \mathbb{P}(h(x) \in \mathcal{C}(x) \mid (x, z) \in \mathcal{F}_N, h(x) \notin \mathcal{Y}) \approx \mathbb{P}(h(x) \in \mathcal{C}(x) \mid h(x) \notin \mathcal{Y}) $$
> These enable us to translate the conformal coverage guarantees from the ground-truth feasible region $\mathcal{F}$ to the feasible set $\mathcal{F}_N$ used in the optimization.
>
> Assumption 4.1 is therefore reasonable when the calibration data adequately covers the parts of the input space that intersect the feasible region $\mathcal{F}_N$, both within the ground-truth feasible region $\mathcal{F}$ and its complement $\mathcal{F}^c$. In this sense, it aligns with standard generalization assumptions that require the training and calibration data to be representative of the regions where predictions are deployed. Alternatively, Assumption 4.1 can be approximated using more granular conditional conformal methods, by partitioning the optimization region into finer subregions and enforcing local coverage guarantees within each, which can then be translated to the feasible set $\mathcal{F}_N$.
>
> We will revise the paper to better explain this interpretation and expand the discussion on when the assumption might fail. For instance, the assumption may break down if the feasible region $\mathcal{F}_N$ is heavily concentrated in areas where the calibration set is sparse or systematically miscalibrated. In our experimental settings, we observe good empirical alignment between target and achieved coverage (Appendix E), suggesting that Assumption 4.1 holds reasonably well in practice in realistic data scenarios.
>
> - Q3: Yes, thank you for raising this point. We agree that our original wording is imprecise.  Assumption 4.1 is meant to apply to the full feasible set of the C-MICL problem, $\mathcal{F}_N = \{(x,z) \in \mathcal{X}: g(x,z)\leq 0, \mathcal{C}(x) \subseteq \mathcal{Y}\}$, rather than only to the containment condition $ \mathcal{C}(x) \subseteq \mathcal{Y}$. We will revise the statement to clarify this and more accurately state the meaning of our assumption as used in the proof. Thanks again for pointing this out.
>
> - W3 + Q4: We thank the reviewer for this important question. However, since $\mathcal{C}(x)$ is defined using $\widehat{h}(x)$ rather than the true function $h(x)$, the feasible regions defined by the oracle constraint $h(x) \in \mathcal{Y}$ and the conformal set containment $\mathcal{C}(x) \subseteq \mathcal{Y}$ differ in no clear way. In particular, neither region necessarily contains the other, and therefore one is not more conservative nor yields larger/smaller optimal values in general.
>
> On the other hand, relative to the learned function $\widehat{h}(x)$, our conformal constraint is indeed conservative since $\widehat{h}(x) \in \mathcal{C}(x)$ by construction, making the containment constraint $\mathcal{C}(x) \subseteq \mathcal{Y}$ more strict than $\widehat{h}(x) \in \mathcal{Y}$. The latter case corresponds exactly to the MICL approach where point predictions are used directly in the optimization problem without uncertainty quantification. We compare C-MICL to this naive method in Figures 2 and 5, showing optimal values 15\% and 1\% smaller for our regression and classification settings, respectively. Crucially, while MICL achieves better optimal values, it systematically violates ground-truth feasibility constraints as shown in Figures 1 and 4. Our method provides probabilistic ground-truth feasibility guarantees while achieving comparable optimal values.
>
> Nevertheless, we agree that comparing to the ground truth problem's optimal objective value (i.e., solving the optimization problem using the true constraint $h(x) \subseteq \mathcal{Y}$) is a relevant baseline. We will add this to the updated version and discuss the corresponding results.

---

> > ### Comment · Reviewer_Pih6 · 2025-08-03
> >
> > Thanks for your rebuttal. I appreciate your response to my question about Assumption 4.1, and I will raise my score accordingly.

---

### Decision · Program_Chairs · 2025-09-17

**Decision:**

Accept (spotlight)

**Comment:**

I would like to thank the authors for submitting an interesting paper on learning constraints from data using a new framework that they call Conformal Mixed-Integer Constraint Learning (C-MICL).  The reviewers are unanimously positive on the paper,  advocating that it provides a new approach for constraint learning from data with probabilistic feasibility guarantees. The paper is well written and provides detailed convincing experiments. Reviewers raised various questions that were overall comprehensively and satisfactorily addressed in the rebuttal.

I would like to recommend acceptance of the paper for NeurIPS, encouraging the authors to include brief summaries of the key discussions with reviewers -- especially additional scholarship putting the paper in the context of prior work, and discussing the conditional independence assumption.

Thank you,
AC.